Journal of Data-centric Machine Learning Research (2026)          Submitted 12/24; Revised 09/25; Published 02/26

# MOMAland: A Set of Benchmarks for Multi-Objective Multi-Agent Reinforcement Learning

**Florian Felten**[1,2,3]                                      FFELTEN@MAVT.ETHZ.CH

**Umut Ucak**[1]                                           UMUT.UCAK.002@STUDENT.UNI.LU

**Hicham Azmani**[4]                                        HICHAM.AZMANI@VUB.BE

**Gao Peng**[5]                                            GAO.PENG@CWI.NL

**Willem Röpke**[4]                                         WILLEM.ROPKE@VUB.BE

**Hendrik Baier**[6,5]                                      H.J.S.BAIER@TUE.NL

**Patrick Mannion**[7]                                      PATRICK.MANNION@UNIVERSITYOFGALWAY.IE

**Diederik M. Roijers**[4,8]                                DIEDERIK.ROIJERS@VUB.BE

**Jordan K. Terry**[2]                                      JKTERRY0@GMAIL.COM

**El-Ghazali Talbi**[9,10]                                  EL-GHAZALI.TALBI@UNIV-LILLE.FR

**Grégoire Danoy**[10,1]                                    GREGOIRE.DANOY@UNI.LU

**Ann Nowé**[4]                                            ANN.NOWE@VUB.BE

**Roxana Rădulescu**[11,4]                                  R.T.RADULESCU@UU.NL

AUTHORS' AFFILIATIONS ARE LISTED IN SECTION 10.

**Reviewed on OpenReview:** HTTPS://OPENREVIEW.NET/FORUM?ID=VZHLRK0SSP

**Editor:** Jakob Nicolaus Foerster

## Abstract

Many challenging tasks such as managing traffic systems, electricity grids, or supply chains involve complex decision-making processes that must balance multiple conflicting objectives and coordinate the actions of various independent decision-makers (DMs). One perspective for formalising and addressing such tasks is multi-objective multi-agent reinforcement learning (MOMARL). MOMARL broadens reinforcement learning (RL) to problems with multiple agents each needing to consider multiple objectives in their learning process. In reinforcement learning research, benchmarks are crucial in facilitating progress, evaluation, and reproducibility. The significance of benchmarks is underscored by the existence of numerous benchmark frameworks developed for various RL paradigms, including single-agent RL (e.g., Gymnasium), multi-agent RL (e.g., PettingZoo), and single-agent multi-objective RL (e.g., MO-Gymnasium). To support the advancement of the MOMARL field, we introduce MOMAland, the first collection of standardised environments for multi-objective multi-agent reinforcement learning. MOMAland addresses the need for comprehensive benchmarking in this emerging field, offering over 10 diverse environments that vary in the number of agents, state representations, reward structures, and utility considerations. To provide strong baselines for future research, MOMAland also includes algorithms capable of learning policies in such settings.[1]

**Keywords:** reinforcement learning, multi-objective optimisation, multi-agent learning, benchmarks, decision-making

---

[1]Documentation: HTTPS://MOMALAND.FARAMA.ORG/.

# 1 Introduction

Often, domains of critical social relevance such as smart electrical grids (Lu et al., 2022), traffic systems (Houli et al., 2010), taxation policy design (Zheng et al., 2022), or infrastructure management planning (Leroy et al., 2023) involve controlling multiple agents while making compromises among several conflicting objectives. The multi-agent aspect of the above-mentioned domains has been well studied, and there is a wealth of literature on multi-agent approaches spanning several decades (Gronauer and Diepold, 2022). Separately, over the last 20 years, there has been an increasing interest in the multi-objective aspect of such domains (Hayes et al., 2022). However, the intersection of these two aspects, multi-objective multi-agent decision making (MOMADM) (Rădulescu, 2021), has not received much attention. Indeed, problems are often simplified by hard-coding a trade-off among objectives or centralising all decisions in a single agent. We argue that many, if not most, complex problems of social relevance have both a multi-objective and a multi-agent dimension. This is because such problems often affect multiple stakeholders, who may care about different aspects of the outcome, and may have different preferences for them. As such, it is crucial to advance the field of MOMADM to enable future progress in the application of artificial intelligence (AI).

The development of standardised benchmarks is a key factor that has driven progress in various areas of AI over the years. Without standardised, publicly available benchmarks, researchers spend a lot of unnecessary time re-implementing test environments from published papers, reproducibility is made much more difficult, and results published in different papers are potentially incomparable (Patterson et al., 2023; Felten et al., 2023). Suites of standardised benchmarks have helped to address these issues already in some fields of AI such as reinforcement learning (RL). Such benchmarks are exemplified by the seminal Gymnasium library (Towers et al., 2023) for single-objective single-agent RL, the PettingZoo library (Terry et al., 2021) for multi-agent RL (MARL), and MO-Gymnasium (Alegre et al., 2022) for multi-objective RL (MORL). Yet, there is no existing library dedicated to multi-objective multi-agent reinforcement learning (MOMARL).

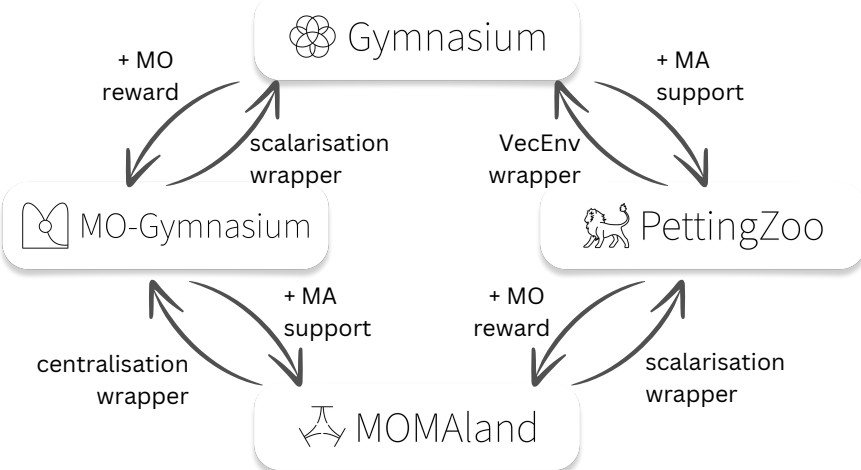

FIGURE 1: Overview of the libraries related to MOMAland within the Farama Foundation.

In this article, we introduce MOMAland, the first publicly available set of MOMARL benchmarks under standardised APIs. MOMAland, incorporated within the Farama Foundation ecosystem (see Figure 1), draws inspiration from analogous projects and currently offers over 10 configurable environments encompassing diverse MOMARL research settings. By embracing open-source principles and inviting contributions, we anticipate that MOMAland will evolve in tandem with research trends and host new environments in the future. As we discuss further in Section 2, MOMAland aims to contribute to the unification of rather fractured evaluation practices, in order to provide researchers with clear and objective data on how well their algorithms perform with respect to other methods.

Additionally, MOMAland includes utilities and learning algorithms intended to establish baselines for future research in MOMARL. Notably, it offers utilities enabling the utilisation of existing MORL and MARL solving methods through centralisation or scalarisation strategies. Importantly, while the provided baselines can find solutions for certain MOMARL settings, MOMAland also features challenges with no known solution concept. Addressing these challenges requires tackling open research questions before deriving appropriate solving methods. Having set this framework, we strongly encourage contributing new work in MOMARL to the MOMAland baselines.

The remainder of this paper is organised as follows: Section 2 describes previous work in this area, Section 3 clarifies background information on the field of MOMARL, Section 4 illustrates the APIs exposed and utilities provided in MOMAland, Section 5 lists the environments currently included in our library, Section 6 presents algorithms designed to address the introduced environments along with baseline results, Section 7 discusses future challenges in this new field of research, and finally, Section 8 concludes our work.

## 2 Related Work

Unlike traditional machine learning settings that often rely on fixed data sets, RL problems typically do not, making replication of experimental results challenging (Felten et al., 2023; Patterson et al., 2023). Indeed, although the Markov Decision Process (MDP) definitions are typically well-specified in research papers, their actual instantiation can be influenced by implementation decisions. Notably, even minor discrepancies in environment specifications can have a substantial impact on RL algorithms' performance. Moreover, re-implementing some of these environments, such as those based on the MuJoCo engine (Todorov et al., 2012), from scratch would require a significant amount of effort for researchers.

To mitigate these issues and accelerate research in standard RL settings, Gymnasium (Towers et al., 2023)—formerly known as OpenAI Gym (Brockman et al., 2016)—introduced a standard API and collection of versioned environments. With millions of downloads, this library has become the standard for RL research. Gymnasium allows researchers to evaluate the performance of their contributions on a varied collection of environments with few code changes, and ensures that the environments used for comparison against state-of-the-art algorithms are the same.

However, Gymnasium is tailored for single-agent, single-objective MDPs and does not offer support for more complex domains involving multiple agents or objectives. Hence, it has been extended in various ways, such as PettingZoo (Terry et al., 2021) or OpenSpiel (Lanctot et al., 2020) for MARL and MO-Gymnasium (Alegre et al., 2022) for MORL. The Farama

Foundation, a recently created nonprofit, takes care of maintaining most of these libraries up to high standards.

Demonstrating the rising interest in settings involving multiple agents and objectives, some initial MOMARL benchmarks were proposed by Ajridi et al. (2023) and Geng et al. (2024). Additionally, Röpke (2022) introduced Ramo[2], a framework offering a collection of algorithms and utilities for solving multi-objective normal-form games which are a particular model studied in MOMARL. However, there is currently no widely adopted library providing reliable and maintained implementations of general MOMARL environments (Hu et al., 2023), and this is precisely the gap targeted by MOMAland.

Finally, over time, numerous RL learning libraries containing algorithms that adhere to standardised APIs have been released. For example, Stable-Baselines3 (Raffin et al., 2021) and cleanRL (Huang et al., 2022) offer a collection of high-quality implementations of state-of-the-art algorithms. Libraries for MARL like EpyMARL (Papoudakis et al., 2021), and for MORL such as MORL-Baselines (Felten et al., 2023) are also available. Nonetheless, the recent development of MOMARL and the absence of standardised environments mean that only a limited number of methods (e.g., MO-MIX from Hu et al., 2023) that can operate in these conditions have been developed, with no dedicated libraries for MOMARL yet. To address this, we also include utilities and baseline algorithms to provide initial solutions to some of the introduced environments.

## 3 Multi-Objective Multi-Agent Reinforcement Learning

In this section, a formal definition and notations of the MOMARL problem are first provided in Section 3.1. Then, solution concepts under different assumptions are discussed in Section 3.2. Following this, metrics for evaluating and contrasting solving methods in this area are presented in Section 3.3.

### 3.1 Formal Definition

The most general framework for modelling multi-objective multi-agent decision-making settings is the multi-objective partially observable stochastic game (MOPOSG). MOPOSGs extend Markov decision processes (Puterman, 1990) to both multiple agents and multiple objectives, under the most general setting in which agents do not observe the full state of the environment (Rădulescu et al., 2020a).

**Definition 1 (Multi-objective partially observable stochastic game)** *A multi-objective partially observable stochastic game is a tuple $M = (\mathcal{S}, \mathcal{A}, T, \boldsymbol{R}, \Omega, \mathcal{O})$, with $n \geq 2$ agents and $d \geq 2$ objectives, where:*

- *$\mathcal{S}$ is the state space;*

- *$\mathcal{A} = \mathcal{A}_1 \times \cdots \times \mathcal{A}_n$ is the set of joint actions, $\mathcal{A}_i$ is the action set of agent $i$;*

- *$T \colon \mathcal{S} \times \mathcal{A} \to \Delta(\mathcal{S})$ represents the probabilistic transition function;*

- *$\boldsymbol{R} = \boldsymbol{R}_1 \times \cdots \times \boldsymbol{R}_n$ are the reward functions, where $\boldsymbol{R}_i \colon \mathcal{S} \times \mathcal{A} \times \mathcal{S} \to \mathbb{R}^d$ is the vectorial reward function of agent $i$ for each of the $d$ objectives;*

---

[2]https://ramo.readthedocs.io/en/latest/

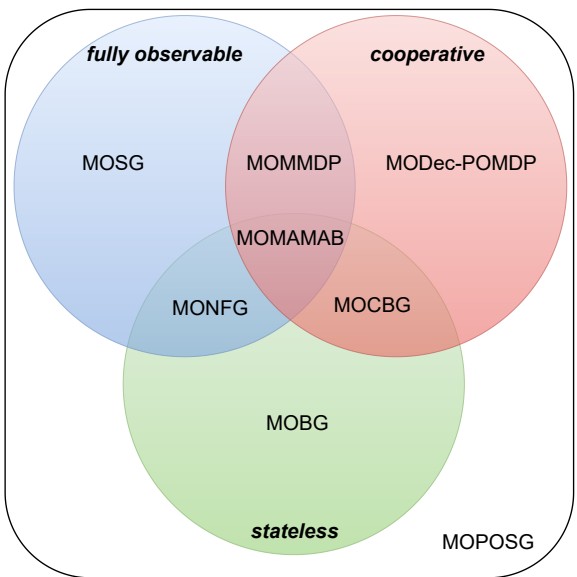

FIGURE 2: Multi-objective multi-agent decision-making models characterised along three axes: (i) observability; (ii) cooperativeness; (iii) statefulness (Rădulescu et al., 2020a).

- $\Omega = \Omega_1 \times \cdots \times \Omega_n$ *is the set of joint observations,* $\Omega_i$ *is the observation set of agent i;*

- $\mathcal{O}: \mathcal{S} \times \mathcal{A} \to \Delta(\Omega)$ *is the observation function, which maps each state—joint action pair to a probability distribution over the joint observation space.*

After every timestep, each agent receives an observation according to the observation function $\mathcal{O}$, instead of directly observing the state. In this case, memory is required for agents to successfully learn in the environment (Spaan, 2012). A particular form of this memory occurs when agents consider the complete history of the current trajectory denoted as $h \in \mathcal{H}$ (Hauskrecht, 2000) (i.e., the complete trace of executed actions and received observations).

By making additional assumptions on the MOPOSG model, regarding observability, the structure of the reward function, or whether the problem is sequential or not, we can derive a subset of models such as the multi-objective stochastic game (MOSG), multi-objective decentralised partially observable Markov decision process (MODec-POMDP), multi-objective Bayesian game (MOBG), multi-objective cooperative Bayesian game (MOCBG), multi-objective multi-agent Markov decision process (MOMMDP), multi-objective normal form game (MONFG), or multi-objective multi-agent multi-armed bandit (MOMAMAB), as illustrated in Figure 2 (Rădulescu et al., 2020a).

In such settings, an agent behaves according to a policy $\pi_i : \mathcal{H} \times \mathcal{A}_i \to [0, 1]$, that provides a probabilistic mapping between an agent's history and its action set. In MOMARL, agents usually aim to optimise their individual expected discounted return obtained from a joint policy $\boldsymbol{\pi}$. Formally,

$$\boldsymbol{v}_i^{\boldsymbol{\pi}} = \mathbb{E}\left[\sum_{t=0}^{\infty} \gamma^t \boldsymbol{R}_i(s_t, \boldsymbol{a}_t, s_{t+1}) \mid \boldsymbol{\pi}\right] \tag{1}$$

where $\boldsymbol{\pi} = (\pi_1, \ldots, \pi_n)$ is the joint policy of the agents acting in the environment, $\gamma$ is the discount factor and $\boldsymbol{R}_i(s_t, \boldsymbol{a}_t, s_{t+1})$ is the vectorial reward obtained by agent $i$ for the joint action $\boldsymbol{a}_t \in \mathcal{A}$ at state $s_t \in \mathcal{S}$.

Note that since an agent only directly controls its own policy $\pi_i$, this introduces subtleties not present in single-agent settings, such as non-stationarity (stemming from agents simultaneously learning in the environment) and additional credit assignment challenges (i.e., identifying the individual contribution of agents to the resulting reward signal). Moreover, as a consequence of the fact that the value functions are vectors, $\boldsymbol{v}_i^{\boldsymbol{\pi}} \in \mathbb{R}^d$, they only offer a partial ordering over the policy space. Determining the optimal policy requires additional information on how agents prioritise the objectives or what their preferences over the objectives are. We can capture such a trade-off choice using a *utility function*, $u_i : \mathbb{R}^d \to \mathbb{R}$, that maps the vector to a scalar value.

In the context of multi-objective multi-agent decision-making, Rădulescu et al. (2020a) propose a taxonomy along the reward and utility axes. Namely, they propose to characterise settings in terms of *individual or team rewards* and *individual, team or social choice utility*. We will use the same dimensions to characterise the solution concepts presented below, as well as the environments introduced by MOMAland.

### 3.2 Solution Concepts

The multi-objective decision-making literature (Roijers et al., 2013b; Hayes et al., 2022) discusses two distinct perspectives for defining solutions in multi-objective settings. We briefly discuss each perspective below, tailored to the multi-objective multi-agent setting.

#### 3.2.1 Axiomatic approach

The axiomatic approach designates the Pareto set (PS) as the optimal solution set, under the minimal assumption that the utility function is a monotonically increasing function. Informally, Pareto dominance introduces a partial ordering over vectors, where one vector is preferred over another when it is at least equal on all objectives and strictly better on at least one. We define this formally in the following definition.

**Definition 2 (Pareto dominance)** *Let $\boldsymbol{v}, \boldsymbol{v}' \in \mathbb{R}^d$. We say $\boldsymbol{v}$ Pareto dominates $\boldsymbol{v}'$, denoted $\boldsymbol{v} \succ_P \boldsymbol{v}'$, whenever*

$$\forall j \in \{1, \ldots, d\} : v_j \geq v_j' \wedge \exists j \in \{1, \ldots, d\} : v_j > v_j'. \tag{2}$$

*When only ensuring that*

$$\forall j \in \{1, \ldots, d\} : v_j \geq v_j', \tag{3}$$

*we refer to this as weak Pareto dominance and denote this by $\boldsymbol{v} \succeq_P \boldsymbol{v}'$.*

**Team reward setting**  When all agents must cooperate, they often share a team reward, i.e. $\boldsymbol{v}_1^{\boldsymbol{\pi}} = \boldsymbol{v}_2^{\boldsymbol{\pi}} = \ldots = \boldsymbol{v}_n^{\boldsymbol{\pi}}$, denoted as $\boldsymbol{v}^{\boldsymbol{\pi}}$. Given this shared reward, Pareto dominance can be straightforwardly applied. We define this below.

**Definition 3 (Pareto dominance for team reward)** *In a team-reward setting, we say that a joint policy $\boldsymbol{\pi}$ Pareto dominates another joint policy $\boldsymbol{\pi}'$ whenever $\boldsymbol{v}^{\boldsymbol{\pi}} \succ_P \boldsymbol{v}^{\boldsymbol{\pi}'}$.*

We subsequently define the set of all joint policies which are not Pareto dominated as the *Pareto set*.

**Definition 4 (Pareto set for team reward)** *Let* $\mathbf{\Pi}$ *be a set of joint policies. The Pareto set* $\mathcal{P}(\mathbf{\Pi})$ *in a team reward setting contains all joint policies that are pairwise undominated, i.e.*

$$\mathcal{P}(\mathbf{\Pi}) = \{\boldsymbol{\pi} \in \mathbf{\Pi} \mid \nexists \, \boldsymbol{\pi}' \in \mathbf{\Pi} : \boldsymbol{v}^{\boldsymbol{\pi}'} \succ_P \boldsymbol{v}^{\boldsymbol{\pi}}\}, \tag{4}$$

The Pareto front (PF), denoted as $\mathcal{F}(\mathcal{P})$,[3] contains the value vectors corresponding to all Pareto optimal policies $\boldsymbol{\pi} \in \mathcal{P}(\mathbf{\Pi})$.[4] It is usually presented to the decision-maker after the learning process to let them choose the desired behaviour to deploy. For instance, Figure 3 illustrates the outcomes of running a MOMARL algorithm (Algorithm 1) in team reward settings. This has been performed in one of our new environments where agents learn to make a formation around a fixed target. In the depiction, the yellow sphere represents the target, while the agents are depicted by the red spheres. This environment offers a clear demonstration of the various behaviours achievable by making different compromises among two objectives involving being close to the target and far from other agents. In this environment, the agents converge to a final position determined by the desired trade-off specified by the decision-maker. Opting for a tight formation around the target enhances the surrounding objective, albeit at the expense of greater collision risks. See Appendix C for further details on the environment.

**Individual reward setting**    While the Pareto set and Pareto front are natural solutions in cooperative settings, extending this to settings where each agent receives a different reward vector is non-trivial. Observe that the value function for joint policies, $\boldsymbol{V}^{\boldsymbol{\pi}} = [\boldsymbol{v}_1^{\boldsymbol{\pi}} \boldsymbol{v}_2^{\boldsymbol{\pi}} \ldots \boldsymbol{v}_n^{\boldsymbol{\pi}}]^\mathsf{T}$, in multi-agent multi-objective agent settings, is a matrix where each row represents the payoff vector of a particular agent. A well-known solution concept extending Pareto dominance to this setting is the *Pareto-Nash equilibrium* (Lozovanu et al., 2005) in which each player's value vector should be in the Pareto front induced by keeping the opponents' policies fixed.

**Definition 5 (Pareto-Nash dominance)** *We say a joint policy* $\boldsymbol{\pi}$ *Pareto-Nash dominates another joint policy* $\boldsymbol{\pi}'$, *denoted as* $\boldsymbol{V}^{\boldsymbol{\pi}} \succ_{PN} \boldsymbol{V}^{\boldsymbol{\pi}'}$, *whenever*

$$\forall i \in \{1, \ldots, n\} : \boldsymbol{v}_i^{\boldsymbol{\pi}} \succeq_P \boldsymbol{v}_i^{\boldsymbol{\pi}'} \wedge \exists i \in \{1, \ldots, n\} : \boldsymbol{v}_i^{\boldsymbol{\pi}} \succ_P \boldsymbol{v}_i^{\boldsymbol{\pi}'}. \tag{5}$$

Note that this definition does not conflict with Definition 3 since in team reward settings each row in the value matrix is equal. The set of all Pareto-Nash equilibria then contains all joint policies which are not dominated.

**Definition 6 (Pareto-Nash set)** *Let* $\mathbf{\Pi}$ *be a set of joint policies. The Pareto-Nash set* $\mathcal{PN}(\mathbf{\Pi})$ *contains all joint policies that are pairwise undominated, i.e.*

$$\mathcal{PN}(\mathbf{\Pi}) = \{\boldsymbol{\pi} \in \mathbf{\Pi} \mid \nexists \, \boldsymbol{\pi}' \in \mathbf{\Pi} : \boldsymbol{V}^{\boldsymbol{\pi}'} \succ_{PN} \boldsymbol{V}^{\boldsymbol{\pi}}\} \tag{6}$$

---

[3]For simplicity, we often use $\mathcal{F}$ to denote the PF, as the expected value vectors are inherently linked to the policies.

[4]The concepts of PS and PF are often used interchangeably in the literature. We make a clear distinction between the two for mathematical rigour.

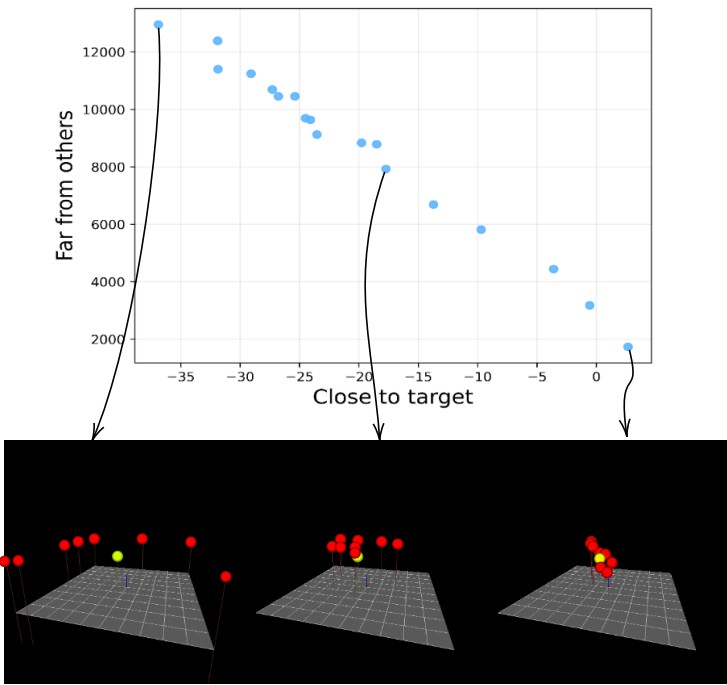

FIGURE 3: Pareto Front and corresponding final positions learned on a CrazyRL environment (introduced below).

We note that there is little work so far on the *individual reward* setting with *unknown utility functions*, so this more general setting remains an important open challenge in MOMARL. Indeed, to the best of our knowledge, there is a limited number of methods capable of identifying a Pareto-Nash set of policies. A notable example, but limited to games with additional structure (e.g., symmetric games) is introduced in the work of Somasundaram and Baras (2009). All other methods either assume a team reward setting, falling back to a team Pareto set, or assume a known utility function, falling back to a Nash equilibrium, which we discuss below.

### 3.2.2 UTILITY-BASED APPROACH

The utility-based approach advocates for exploiting any additional domain knowledge that might be available regarding the user's utility function. Such additional knowledge can lead to smaller optimal sets (e.g., if the utility function is known to be linear), or less time spent on exploring regions of the objective space that are not of interest to the user (e.g., when the user requires some minimum value for a certain objective). When no additional knowledge on the utility function is available, the utility-based approach falls back on the axiomatic approach.

Roijers et al. (2013b) define two optimisation criteria in multi-objective decision-making when applying the utility function to the vector-valued outcomes. One can compute the expected value of the payoffs of a policy first and then apply the utility function, leading to the *scalarised expected returns (SER)* optimisation criterion:

$$v_{u_i}^{\boldsymbol{\pi}} = u_i \left( \mathbb{E} \left[ \sum_{t=0}^{\infty} \gamma^t \boldsymbol{R}_i(s_t, \mathbf{a}_t, s_{t+1}) \mid \boldsymbol{\pi} \right] \right) \tag{7}$$

where $v_{u_i}^{\boldsymbol{\pi}}$ is the scalarised return derived by agent $i$. Alternatively, under the *expected scalarised returns (ESR)* optimisation criterion (Hayes et al., 2021; Reymond et al., 2023), the utility function is applied before computing the expectation:

$$v_{u_i}^{\boldsymbol{\pi}} = \mathbb{E} \left[ u_i \left( \sum_{t=0}^{\infty} \gamma^t \boldsymbol{R}_i(s_t, \mathbf{a}_t, s_{t+1}) \right) \mid \boldsymbol{\pi} \right] \tag{8}$$

Semantically, the two optimisation criteria distinguish between settings in which users are interested in optimising the utility over multiple policy executions (SER), or over each policy application (ESR).

The majority of work on multi-objective multi-agent decision-making so far has taken a utility-based perspective, assuming that each agent has some known utility function $u_i$, dictating the preferred trade-off among the objectives (Rădulescu et al., 2020b, 2021; Röpke et al., 2022), and has mostly focused on stateless settings (i.e., MONFGs).

For the scope of this work, we introduce below one of the solution concepts identified for the individual utility setting, namely the Nash equilibrium (Nash, 1950). We define $\boldsymbol{\pi}_{-i} = (\pi_1, \ldots, \pi_{i-1}, \pi_{i+1}, \ldots, \pi_n)$ to be a strategy profile without player's $i$ strategy. We can thus write $\boldsymbol{\pi} = (\pi_i, \boldsymbol{\pi}_{-i})$.

**Definition 7 (Nash equilibrium)** *A joint policy $\boldsymbol{\pi}^{NE}$ is a Nash equilibrium if, for each agent $i \in \{1, ..., n\}$ and for any alternative policy $\pi_i$, no agent can improve its scalarised return by unilaterally changing its policy:*

$$v_{u_i}^{(\pi_i^{NE}, \boldsymbol{\pi}_{-i}^{NE})} \geq v_{u_i}^{(\pi_i, \boldsymbol{\pi}_{-i}^{NE})}. \tag{9}$$

For a detailed discussion on each of the MOMARL taxonomy settings and solution concepts, we refer to Rădulescu et al. (2020a).

### 3.3 Evaluation of MOMARL algorithms

Because of the additional complexity in MOMARL compared to single-agent, single-objective RL, solution concepts vary, leading to different evaluation methods for MOMARL algorithms. In this section, we present commonly used evaluation methods in both MORL and MARL domains and examine their suitability for MOMARL settings. First, we outline performance indicators for settings where the agents' utilities are unknown, followed by a discussion on settings where the utilities are known.

#### 3.3.1 Performance Indicators for Unknown Utility

First, it is important to acknowledge that, due to the scarcity of research in general settings, there are few, if any, established methods for evaluating the effectiveness of approaches that identify a Pareto-Nash set. Nevertheless, as discussed above, when the agents' utilities remain unknown and under the team reward setting, the Pareto set and Pareto front (Definition 4)

are usually designated as optimal solution sets. These solution concepts have been well studied in multi-objective optimisation literature, as well as in MORL more recently.

Compared to single-objective RL, the assessment and comparison of PFs obtained by different algorithms pose challenges due to the PFs being collections of points, i.e. there is no existing ordering of PFs. Defining such an order is not straightforward for two main reasons. Firstly, Pareto fronts discovered by various algorithms can be intertwined, meaning that one algorithm may outperform another in a portion of the objective space while the opposite may hold true in another portion. Secondly, PFs in high dimensions present difficulty in visualisation. In practice, *performance indicators* become useful to transform a PF into a scalar value. This establishes an order among PFs and enables comparisons. Various types of performance indicators have been introduced in the MO literature for this purpose. However, it is important to note that compressing a set of points into a single scalar value inevitably introduces bias. Hence, multiple performance indicators (assessing different criteria) are often used in practice when comparing PFs. These criteria include *convergence*, which assesses how close to optimality the discovered policies are, and *diversity*, which evaluates the variety of compromises the discovered policies offer to the user.

Similar to solution concepts, performance indicators can be categorised into two groups: *axiomatic indicators*, which do not make any assumptions about the decision maker's (DM's) utility, and *utility-based indicators*, which assume specific restrictions on the DM's utility function, e.g. linearity. Several of these indicators, employed throughout this work, are given below.

**Cardinality.** This metric is computed by considering the number of points within the approximated PF found by the algorithm ($\tilde{\mathcal{F}}$). It offers insights into the diversity of $\tilde{\mathcal{F}}$ by indicating the number of trade-offs identified.

$$\mathrm{C}(\tilde{\mathcal{F}}) = |\tilde{\mathcal{F}}|.$$

**Hypervolume.** This is a hybrid metric quantifying both a PF's convergence and diversity. Given an approximate PF, $\tilde{\mathcal{F}}$, and a pessimistic reference point, $\boldsymbol{z}_{\mathrm{ref}}$, the hypervolume indicator represents the volume of the objective space starting from $\boldsymbol{z}_{\mathrm{ref}}$ that is weakly dominated by $\tilde{\mathcal{F}}$. Formally, the hypervolume metric (Zitzler, 1999) is defined as:

$$\mathrm{HV}(\tilde{\mathcal{F}}, \boldsymbol{z}_{\mathrm{ref}}) = \Lambda \left( \bigcup_{\substack{\boldsymbol{v}^{\boldsymbol{\pi}} \in \tilde{\mathcal{F}} \\ \boldsymbol{v}^{\boldsymbol{\pi}} \succeq_P \boldsymbol{z}_{\mathrm{ref}}}} \mathrm{Box}(\boldsymbol{v}^{\boldsymbol{\pi}}, \boldsymbol{z}_{\mathrm{ref}}) \right),$$

where $\Lambda(\cdot)$ is the Lebesgue measure and $\mathrm{Box}(\boldsymbol{v}^{\boldsymbol{\pi}}, \boldsymbol{z}_{\mathrm{ref}}) = \{\boldsymbol{p} \in \mathbb{R}^d \mid \boldsymbol{v}^{\boldsymbol{\pi}} \succeq_P \boldsymbol{p} \succeq_P \boldsymbol{z}_{\mathrm{ref}}\}$ denotes the box delimited above by $\boldsymbol{v}^{\boldsymbol{\pi}} \in \tilde{\mathcal{F}}$ and below by $\boldsymbol{z}_{\mathrm{ref}}$. The reference point used in the hypervolume computation is typically an estimate of the worst-possible value per objective.

**Expected utility.** In the case where the utility function of the DM, $u$, is linear, it becomes feasible to represent the expected utility over a distribution of reward weights, $\mathcal{W}$, using the expected utility (EU) metric (Zintgraf et al., 2015). The EU metric is then defined as:

$$\mathrm{EU}(\tilde{\mathcal{F}}) = \mathbb{E}_{\boldsymbol{w} \sim \mathcal{W}} \left[ \max_{\boldsymbol{v}^{\boldsymbol{\pi}} \in \tilde{\mathcal{F}}} \boldsymbol{v}^{\boldsymbol{\pi}} \cdot \boldsymbol{w} \right].$$

### 3.3.2 Performance Indicators for Known Utility

Another setting for evaluating MOMARL algorithms is when the utility of the agents is known *a priori*. This allows falling back to single-objective MARL solution concepts and using the indicators provided in this field. A few examples of how one can evaluate the performance of agents in this case, depending on the nature of the task at hand include learning curves depicting achieved individual or joint utility over the learning process; analysing the cooperation or coordination capacity of the learned policies (Gorsane et al., 2022); using game theoretic concepts such as convergence to social optimum (i.e., outcome maximising population welfare, Nash equilibria (Röpke et al., 2022), correlated equilibria (Rădulescu et al., 2020b), or cyclic equilibria (Röpke et al., 2022)).

## 4 APIs and Utilities

MOMAland extends both PettingZoo APIs by returning a vectorial reward, i.e., a NumPy array (Harris et al., 2020), instead of a scalar for each agent.

```python
from momaland.envs.multiwalker_stability import momultiwalker_stability_v0 as _env

env = _env.parallel_env(render_mode="human")
observations, infos = env.reset(seed=42)
while env.unwrapped.agents:
    actions = {agent: policies[agent](observations[agent]) for agent in env.unwrapped.agents}

    # vec_reward is a dict[str, numpy array]
    observations, vec_rewards, terminations, truncations, infos = env.step(actions)

env.close()
```

Listing 1: Parallel API usage.

The first API, referred to as *parallel*, enables all agents to act simultaneously, as demonstrated in Listing 1. In this mode, signals such as observations, rewards, terminations, truncations, and additional information are consolidated into dictionaries, mapping agent IDs to their respective signals (line 9). Similarly, all actions are provided simultaneously to the step function as a dictionary, mapping each agent's ID to its corresponding action (line 6).

The second API, termed *agent-environment cycle* (AEC), is suitable for turn-based scenarios, such as board games (Terry et al., 2021). A typical usage of this API is depicted in Listing 2. In this setup, each loop provides information solely for the agent currently taking its turn (line 7). For additional notes on the APIs, we refer to the documentation website: https://momaland.farama.org/api/aec/.

```
1  from momaland.envs.multiwalker_stability import momultiwalker_stability_v0 as
   ↪  _env
2
3  env = _env.env(render_mode="human")
4  env.reset(seed=42)
5  for agent in env.agent_iter():
6      # vec_reward is a numpy array
7      observation, vec_reward, termination, truncation, info = env.last()
8      if termination or truncation:
9          action = None
10     else:
11         action = policies[agent](observation)
12     env.step(action)
13 env.close()
```

LISTING 2: AEC API usage.

These APIs enable modelling all our benchmarking environments and offer the advantage of aligning closely with PettingZoo's conventions, thus facilitating comprehension for MARL practitioners and reuse of existing utilities such as SuperSuit's wrappers Terry et al. (2020). Additionally, MOMALAND provides utilities to expose most environments through both APIs (with the exception of some board games, where support for the parallel API is deemed unnecessary).

## 4.1 Utilities

In addition to environments and standard APIs, MOMAland provides several utilities that help algorithm designers in creating and evaluating algorithms in the proposed environments.

The library offers wrappers (Table 1) that allow modifying one aspect of the environment, such as normalising observations. Importantly, MOMALAND environments are compatible with PettingZoo and SuperSuit wrappers, as long as they do not alter the reward vectors. This allows relying on stable implementations and avoiding code duplication. Nevertheless,

| Wrapper | Resulting setting | Reward | Utility | Obs. space | Act. space |
|---------|-------------------|--------|---------|------------|------------|
| Centralisation wrapper (*CentraliseAgent*) | MORL | Team | Team Any | c/d | c/d |
| Scalarisation wrapper (*LineariseReward*) | MARL | Any | Individual Linear | c/d | c/d |
| Normalisation wrapper (*NormaliseReward*) | MOMARL | Any | - | c/d | c/d |

TABLE 1: Overview of wrappers implemented in MOMAland.

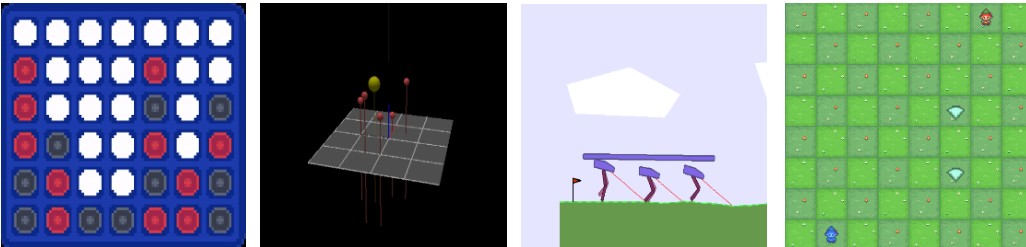

FIGURE 4: Visualization of some environments in MOMALAND. From left to right: MO-Connect4, CrazyRL/Surround, MO-MultiWalker-Stability, MO-ItemGathering.

MOMALAND provides wrappers dedicated to handling the vectorial rewards, as this is the main difference with PettingZoo:

- The *CentraliseAgent* wrapper compresses the multi-agent dimension into a single centralised agent, providing direct conversion to the MO-Gymnasium API (Alegre et al., 2022). This adaptation enables learning using multi-objective single-agent algorithms, such as those featured in MORL-Baselines (Felten et al., 2023).

- The *LineariseReward* wrapper enables the transformation of agent reward vectors into scalar values through a weighted sum of reward components, thereby converting multi-objective environments into single-objective ones under the standard PettingZoo API, see Figure 1. This adaptation allows for the utilisation of existing multi-agent RL algorithms to learn for a designated trade-off.

- The *NormaliseReward(idx, agent)* wrapper facilitates the normalisation of the $idx^{\text{th}}$ immediate reward component for a specified agent.

Additionally, MOMALAND includes a set of baseline algorithms showing example usage of the API and previously discussed utilities. These baselines are discussed in more detail in Section 6.

## 5 Environments

MOMALAND provides a variety of environments which offer a diverse range of challenges to benchmark MOMARL algorithms. Table 2 shows an overview of all environments, according to the criteria depicted in Figure 2, which describe all multi-objective multi-agent settings. Our environments cover a spectrum of features, including discrete and continuous state and action spaces, stateless and stateful environments, cooperative and competitive settings, as well as fully and partially observable states. Notably, the current set of environments provided within MOMALAND covers all configurations depicted in Figure 2, except MOBG and MOCBG. Some environments are multi-objective extensions of PettingZoo domains, others have been implemented from the current literature in MOMARL, and some are introduced in this work, e.g. the CrazyRL variants. In the following, we briefly outline each environment; see Appendix C for more details.

**Multi-Objective Beach Problem Domain (MO-BPD)** The Multi-Objective Beach Problem Domain (MO-BPD) (Mannion et al., 2018) is a setting with two objectives, reflecting

| Domain | # of agents | # of objectives | stochastic transitions? | full observability possible? | partial observability possible? | team rewards possible? | individual rewards possible? | discrete/continuous state (d/c) | discrete/continuous actions (d/c) |
|---|---|---|---|---|---|---|---|---|---|
| MO-BPD | 2-$n$ | 2 | ✗* | ✗ | ✓ | ✓ | ✓ | d | d |
| MO-ItemGathering | 2-$n$ | 2-$d$ | ✓ | ✓ | ✗ | ✓ | ✓ | d | d |
| MO-GemMining | 2-$n$ | 2-$d$ | ✗* | - | - | ✓ | ✗ | - | d |
| MO-RouteChoice | 2-$n$ | 2 | ✗* | - | - | ✗ | ✓ | - | d |
| MO-PistonBall | 2-$n$ | 2 | ✗* | ✗ | ✓ | ✓ | ✓ | c | d/c |
| MO-MW-Stability | 2-$n$ | 2 | ✗ | ✓ | ✓ | ✓ | ✓ | c | c |
| CrazyRL/Surround | 2-$n$ | 2 | ✗ | ✓ | ✗ | ✓ | ✓ | c | c |
| CrazyRL/Escort | 2-$n$ | 2 | ✗ | ✓ | ✗ | ✓ | ✓ | c | c |
| CrazyRL/Catch | 2-$n$ | 2 | ✓ | ✓ | ✗ | ✓ | ✓ | c | c |
| MO-Breakthrough | 2 | 1-4 | ✗ | ✓ | ✗ | ✗ | ✓ | d | d |
| MO-Connect4 | 2 | 2-20 | ✗ | ✓ | ✗ | ✗ | ✓ | d | d |
| MO-Ingenious | 2-6 | 2-6 | ✗ | ✓ | ✓ | ✓ | ✓ | d | d |
| MO-SameGame | 1-5 | 2-10 | ✗* | ✓ | ✗ | ✓ | ✓ | d | d |

TABLE 2: Overview of MOMALAND environments. State observability and discreteness are not specified for MO-GemMining and MO-RouteChoice as these are stateless domains. Upper limits specified as $n$ or $d$ signal that the environment in question does not enforce an upper limit on the number of agents or objectives, respectively. (*) These environments can have randomised starting states, but otherwise no stochastic transitions.

the enjoyment of tourists (agents) on their respective beach sections in terms of crowdedness and diversity of attendees. Each beach section is characterised by a capacity and each agent is characterised by a type. These properties, together with the location selected by the agents on the beach sections, determine the vectorial reward received by agents. The number of agents is configurable.

The MO-BPD domain has two reward modes: (i) *individual reward*, where each agent receives the reward signal associated with its respective beach section; and (ii) *team reward*, where the reward signal for each agent is an objective-wise sum over all the beach sections. In terms of mathematical frameworks, under the individual reward setting, the MO-BDP is a MOPOSG, while the team reward setting casts the problem as a MODec-POMDP. This environment models a multi-objective resource selection congestion problem, where agents face the challenge of (indirectly) coordinating their selected resource in order to achieve high outcomes for each objective.

**MO-ItemGathering**   The Multi-Objective Item Gathering domain (Figure 4, rightmost picture), adapted from Källström and Heintz (2019), is a multi-agent grid world, containing items of different colours. Each colour represents a different objective and the goal of the agents is to collect as many objects as possible. The environment is fully configurable in terms of grid size, number of agents, and number of objectives.

MO-ItemGathering is fully observable and has two reward modes: individual rewards (MOSG), where agents are rewarded only for their own collected items, or team rewards (MOMMDP), where agents receive a reward for any object collected by the group. The environment can also accommodate stochasticity, in the form of a probabilistic realisation of the item collection. This environment is a traditional RL grid setting, where agents need to learn time-sensitive shortest path navigation, with the added challenge of discovering all the potential trade-offs for picking the available items.

**MO-GemMining**   In Multi-Objective Gem Mining, extending Gem Mining / Mining Day (Bargiacchi et al., 2018; Roijers et al., 2015; Roijers, 2016) to multiple objectives, a number of villages (agents) send workers to extract gems from different mines. Each gem type represents a different objective. There are restrictions on which mines can be reached from each village. Furthermore, workers influence each other in their productivity. The number of different gem types, villages, and workers per village are configurable.

MO-GemMining is stateless; each action corresponds to one independent mining day. It is fully cooperative and can be modelled as a multi-objective multi-agent multi-armed bandit (MOMAMAB). The main challenge in MO-GemMining is the coordination problem between possibly large numbers of agents. As the number of agents, as well as their interdependency (or more specifically, the induced width of the underlying coordination graph), can be set by the user, MO-GemMining can be used to test for scalability in the number of agents and the complexity of the coordination problem.

**MO-RouteChoice**   MO-RouteChoice is a multi-objective extension of the route choice problem (Thomasini et al., 2023), where a number of self-interested drivers (agents) must navigate a road network. Each driver chooses a route from a source to a destination while minimising two objectives: travel time and monetary cost. Both objectives are affected by the selected routes of the other agents, as the more agents travel on the same path, the higher the associated travel time and monetary cost. The number of agents is configurable. The environment contains various road networks from the original route choice problem (Ramos et al., 2020b; Thomasini et al., 2023), including the Braess's paradox (Braess, 1968) and networks inspired by real-world cities.

MO-RouteChoice is a stateless environment, thus a MONFG, where each agent chooses one of the possible routes from its source to its destination and receives an individual reward based on the joint strategy of all agents. MO-RouteChoice is a congestion game, a setting where the main challenge arises from the conflict between individual agent rationality and the overall social optimum. It is particularly well-suited for studying how multi-objective learning algorithms perform in large-scale, non-cooperative settings, especially in scenarios like the Braess's paradox where selfish, utility-maximizing behavior can lead to counter-intuitive, system-wide inefficiencies.

**MO-PistonBall**   MO-PistonBall is based on an environment published in PettingZoo (Terry et al., 2021) where the goal is to move a ball to the edge of the window by operating several

pistons (agents). This environment supports continuous observations and both discrete and continuous actions. In the original environment, the reward function is individual per piston and computed as a linear combination of two components. Concretely, the total reward consists of a global reward proportional to the distance to the wall and a per-timestep penalty. In the MOMAland adaptation, the environment dynamics are kept unchanged, but now each reward component is returned as an individual objective. The number of agents is configurable.

This environment is a MOPOSG, where the only stochastic transition dynamics occur when determining the initial state of the ball.

**MO-MW-Stability**   Multi-Objective Multi Walker Stability (Figure 4, third picture from the left) is another adaptation of a PettingZoo environment, originally published in Gupta et al. (2017), to multi-objective settings. In this environment, multiple walker agents aim to carry a package to the right side of the screen without falling. This environment also supports continuous observations and actions. The multi-objective version of this environment includes an additional objective to keep the package as steady as possible while moving it. Naturally, achieving higher speed entails greater shaking of the package, resulting in conflicting objectives. The number of agents is configurable.

This environment is cooperative and agents only have a partial view of the global state. Hence, it is a MODec-POMDP. This environment requires a good amount of coordination between the agents to avoid making the package fall, making them a natural fit for studying team reward settings.

**CrazyRL**   CrazyRL (Figure 4, second picture from the left) consists of 3 novel continuous 3D environments in which drones (agents) aim to surround a potentially moving target. The two objectives of the drones are to minimise their distance to the target while maximising the distance between each other. The 3 environments differ in the behaviour of the target, which can be static, move linearly, or actively try to escape the agents.

These environments are cooperative, and agents can perceive the location of everyone else. Hence, they are all MOMMDPs. These environments are notably useful as the tightness of the drones' formation can be explicitly controlled by choosing a different tradeoff; see Fig. 3. They have also been shown to work for real-time control of drones in the real world (CrazyFlies).[5]

The remaining environments are multi-objective adaptations of well-known board and puzzle games played by humans. Games of this type have historically been a popular benchmark for AI due to their challenges for learning, deep planning, and strategic interaction (Campbell et al., 2002; Schrittwieser et al., 2020). The series of environments we propose here add to these challenges with multiple rewards to be traded off against each other over longer sequences of discrete decisions for one or multiple competing agents.

**MO-Breakthrough**   MO-Breakthrough is a multi-objective variant of the two-player, single-objective turn-based board game Breakthrough. In MO-Breakthrough there are still two agents, but up to three objectives in addition to winning: a second objective that

---

[5]https://github.com/ffelten/CrazyRL

incentivises faster wins, a third one for capturing opponent pieces, and a fourth one for avoiding the capture of the agent's own pieces. The board size is configurable as well.

As the game is competitive and fully observable, MO-Breakthrough falls into the category of MOSGs. It allows for the study of highly complex competitive strategies in a scalable discrete environment.

**MO-Connect4**    MO-Connect4 is a multi-objective variant of the two-player, single-objective turn-based board game Connect 4 (Figure 4, leftmost picture). In addition to winning, MO-Connect4 extends this game with a second objective that incentivises faster wins, and optionally one additional objective for each column of the board that incentivises having more tokens than the opponent in that column. As the board size is configurable, so is the number of these objectives.

MO-Connect4 is competitive and fully observable and therefore a MOSG. The game allows for the study of complex competitive strategies in a discrete environment that is scalable in both size and number of objectives.

**MO-Ingenious**    MO-Ingenious is a multi-objective adaptation of the zero-sum, turn-based board game Ingenious. The game's original rules support 2-4 players collecting scores in multiple colours (objectives), with the goal of winning by maximising the minimum score over all colours. In MO-Ingenious, we leave the utility wrapper up to the users and only return the vector of scores in each colour objective. The number of agents, objectives, and board size in MO-Ingenious are configurable.

MO-Ingenious has two reward modes: (i) *individual reward*, where each agent receives scores only for their own actions; and (ii) *team reward*, where all collected scores are shared by all agents. Furthermore, it can be played with (i) *partial observability* as the original game, or in a (ii) *fully observable* mode. In terms of mathematical frameworks, this environment is therefore a MOPOSG, which can be configured to become a MODec-POMDP when playing in team reward mode, a MOSG when playing in fully observable mode, or a MOMMDP when using both. In addition to the challenges inherent in the other discrete games, MO-Ingenious offers the most flexibility with regard to choosing reward structure and observability.

**MO-SameGame**    MO-SameGame is a multi-objective, multi-agent variant of the single-player, single-objective turn-based puzzle game called SameGame. All legal moves in the game remove a group of tokens of the same colour from the board. The original game rewards the player for each action with a number of points that is quadratic in the size of the removed group. MO-SameGame extends this to a configurable number of agents, acting alternatingly, and a configurable number of different types of colours (objectives) to be collected.

MO-SameGame has two reward modes: (i) *individual reward*, where each agent receives points only for their own actions; and (ii) *team reward*, where all collected points are shared by all agents. It is fully observable and can therefore be modelled as a MOSG in individual reward mode, or a MOMMDP when using team rewards. Among the discrete games, this environment offers the most flexibility with regard to the number of agents.

## 6  Baselines

After introducing our collection of challenging environments and utilities, this section demonstrates typical learning results derived from the solution concepts and metrics presented

| Algorithm | Single or multi compromises | Reward | Utility | Obs. space | Act. space |
|---|---|---|---|---|---|
| MOMAPPO | Multi | Team | Team Linear | c/d | c/d |
| Scalarized IQL (Mannion et al., 2018) | Single | Individual | Individual Linear | d | d |

Table 3: Baseline algorithms implemented in MOMAland.

earlier. We provide baselines that allow learning under different settings. These baselines are listed in Table 3. The second column describes whether the algorithm aims at learning one or multiple policies associated with different trade-offs within the multi-objective dimension. The third and fourth columns refer to the classification made in the work of Rădulescu et al. (2020a) and discussed in Section 3.2. It is worth noting that these algorithms do not aim for maximum efficiency, but provide a solid foundation for future work.

Next to these baselines, we also demonstrate the use of the *CentraliseAgent* wrapper for ensuring compatibility of MOMAland with MORL methods. The pipeline for achieving this is thus by reducing the setting to a multi-objective single-agent problem. As MORL methods, we use GPI-LS (Alegre et al., 2023) and PCN (Reymond et al., 2022), which are among the current state of the art in MORL. Our baselines are intended to be accessible to both MORL and MARL practitioners, while also introducing them to the specific challenges of MOMARL. The rest of this section illustrates results obtained by using the algorithms on some of the proposed environments.

## 6.1 Team Reward with Unknown Team Utility

As explained earlier, this setting aims at finding the same solution concepts as single-agent multi-objective RL, i.e., a Pareto set of policies and its linked Pareto front.

### 6.1.1 Solving MOMARL Problems Using Decomposition

Algorithm 1 describes a simple extension of the MAPPO algorithm (Yu et al., 2022) to return a Pareto set of multi-agent policies in cooperative problems. Similar to the works of Felten et al. (2022, 2024), it employs the decomposition technique to divide the multi-objective problem into a collection of single-objective problems which can then be solved by a multi-agent RL algorithm. In this context, a scalarisation function, parameterised by weight vectors, allows performing the decomposition and targeting various areas of the objective space. The most common scalarisation function, weighted sum, is used in this algorithm for its simplicity (through our *LineariseReward* wrapper, line 6). Notice that the rewards of the environment are first normalised to mitigate the difference in scale of each objective (line 5). The weight vectors can be generated using various techniques from MORL and MO optimisation, e.g., optimistic linear support (OLS) (Roijers and Whiteson,

---

**Algorithm 1** MOMAPPO using Decomposition

---

**Input:** Number of weight vector candidates $n$, stopping criterion per weight *stop*, Environment *MOMAenv*.

**Output:** A Pareto set of joint policies $\mathcal{P}$.

1: $\mathcal{P} = \emptyset$
2: $\mathcal{F} = \emptyset$
3: **for** $i \in \{1, \dots, n\}$ **do**
4:      $\boldsymbol{w} = \text{GenerateWeights}(\mathcal{F})$
5:      $NormEnv = \text{NormalizeRewards}(MOMAenv)$
6:      $MAEnv = \text{LinearizeRewards}(NormEnv, \boldsymbol{w})$
7:      $\boldsymbol{\pi} = \text{MAPPO}(MAEnv, stop)$
8:      $\tilde{\boldsymbol{v}}^{\boldsymbol{\pi}} = \text{EvaluatePolicy}(MOMAenv, \boldsymbol{\pi})$
9:      Add $\boldsymbol{\pi}$ to $\mathcal{P}$ and $\tilde{\boldsymbol{v}}^{\boldsymbol{\pi}}$ to $\mathcal{F}$ if $\tilde{\boldsymbol{v}}^{\boldsymbol{\pi}}$ non-dominated in $\mathcal{F}$
10: **end for**
11: **return** $\mathcal{P}$

---

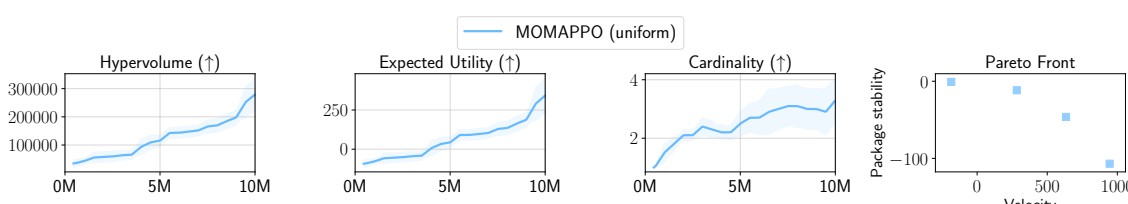

FIGURE 5: Average and 95% confidence intervals of multi-objective performance indicators on training results from MOMAPPO with 20 uniform weights on *mo_ multiwalker_ stability_ v0*. The Pareto Front plot has been extracted from the run with the largest hypervolume.

2017), GPI-LS (Alegre et al., 2023), uniformly (Das and Dennis, 2000; Blank et al., 2021), or randomly (line 4). After training a multi-agent policy for a given trade-off using MAPPO (Yu et al., 2022), the policy is evaluated on the original environment, allowing to compute an estimate of $\boldsymbol{v}^{\boldsymbol{\pi}}$ (line 8) and add the policy to the Pareto set of policies if it is non-dominated (line 9). Finally, the algorithm returns all non-dominated multi-agent policies (line 11).

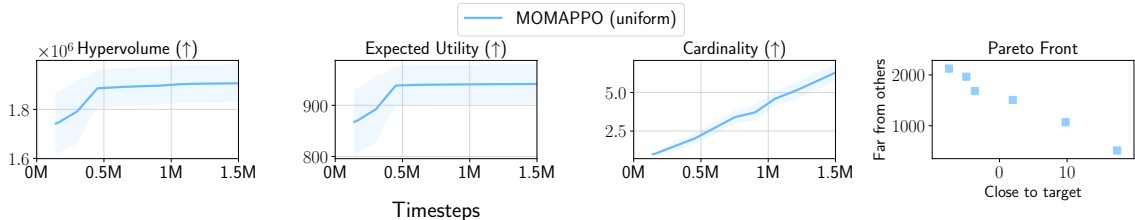

FIGURE 6: Average and 95% confidence intervals of multi-objective performance indicators on training results from MOMAPPO with 10 uniform weights on *catch_ v0*. The Pareto Front plot has been extracted from the run with the largest hypervolume.

Figs. 5 and 6 illustrate the typical metrics results that can be obtained by running MOMAPPO (Algorithm 1) on cooperative environments, *mo_multiwalker_stability_v0* and *catch_v0* in this case. For these runs, the algorithm generated uniformly distributed weight vectors to explore the objective space, training sequentially for each one. Steeper MO learning curves could be obtained by training for different weights in parallel, but complexify the practical implementation. More details on experimental settings are available in Appendix B. The performance indicators plotted have been averaged, and the 95% confidence interval is represented by the shaded area. These reflect the general performance of the algorithm over random seeds ranging from 0 and 9. Moreover, the PF plot gives an idea of the final result for a given run. The reference point used for hypervolume calculation are $[-300, -300]$ and $[-1000, -10]$.

The first thing to notice in the plots is that, on average, this algorithm is able to improve its PF over the training process. Indeed, all indicators improve over the training course. The PF plots reveals that 4 non-dominated policies out of 20 weight vectors have been identified for *multiwalker* and 6 out of 10 for *catch*. It is worth noting that this algorithm is a straightforward adaptation of MARL and MORL techniques. It can be improved by including techniques coming from existing MORL works, such as cooperation between single-objective subproblems, e.g., conditioned networks (Abels et al., 2019) or transfer (Natarajan and Tadepalli, 2005), or more advanced weight vector generation method, such as OLS (Roijers and Whiteson, 2017). A thorough review of such techniques in the context of single-agent MORL is given in the work of Felten et al. (2024).

### 6.1.2 Solving MOMARL Problems Using Centralisation

As mentioned in Section 4.1, MOMAland also provides a *CentraliseAgent* wrapper that turns a multi-agent multi-objective environment into a single-agent multi-objective environment by providing a centralised observation as well as a single vectorial reward signal. The composition method of the vectorial reward is determined by a parameter and can be either a component-wise sum or average of the individual agent rewards. This allows the direct application of methods featured in MORL-Baselines (Felten et al., 2023).

To illustrate the compatibility between MOMAland environments using the *CentraliseAgent* wrapper and MORL-Baselines, we select two approaches, that make different assumptions regarding the environment or utility characteristics. Pareto Conditioned Networks (PCN) (Reymond et al., 2022) is a multi-policy approach designed for deterministic environments. PCN will return an approximate Pareto front as a solution. On the other hand, Generalised Policy Improvement Linear Support (GPI-LS) (Alegre et al., 2023) assumes the utility function is linear and will thus return the convex hull as a solution Hayes et al. (2022).

We present in Figure 7 the results obtained by GPI-LS and PCN on the *moitem_gathering_v0* environment. The experiments are run on the default map of the environment, namely an $8 \times 8$ grid, with 2 agents and 3 different object types (i.e., 3 objectives). The centralised vectorial reward signal is obtained using a component-wise addition over all agents' rewards. The number of timesteps is set to 50 and the results are averaged over 5 runs (random seeds ranging from 40 to 44), with the shaded area representing the 95% confidence interval. The reference point for the hypervolume calculation is $[0, 0, 0]$. More experimental details are available in Appendix B.

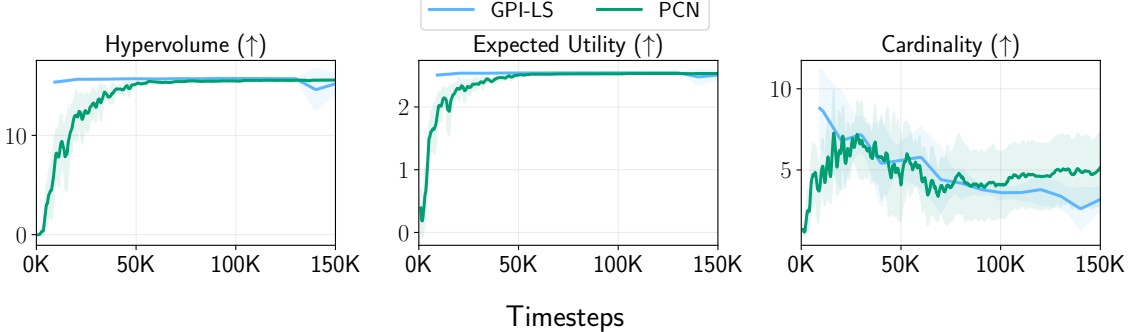

FIGURE 7: Average and 95% confidence intervals of multi-objective performance indicators on training results from GPI-LS and PCN on *mo_item_gathering_v0*, using the centralised agent wrapper.

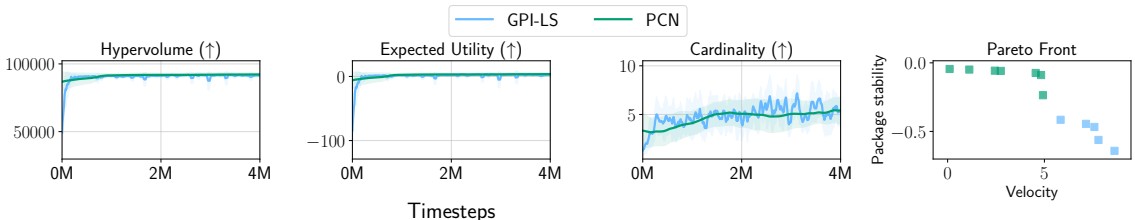

FIGURE 8: Average and 95% confidence intervals of multi-objective performance indicators on training results from GPI-LS and PCN on *mo_multiwalker_stability_v0*, using the centralised agent wrapper.

We observe that for this instance of the MO-ItemGathering environment, both PCN and GPI-LS show consistent learning behaviour over the runs, reaching similar performance in terms of hypervolume and expected utility. In terms of cardinality (i.e., number of solutions in the identified solution set), PCN manages to identify on average one additional solution, in comparison to GPI-LS.

Figure 8 presents the results of GPI-LS and PCN on the *mo_multiwalker_stability_v0* environment, with the default settings of 3 walkers and 2 objectives. The centralised vectorial reward signal is obtained using a component-wise addition over all agents' rewards. The results are averaged over 5 runs (seeds ranging from 40 to 44), with the shaded area representing the 95% confidence interval. The reference point for the hypervolume calculation is $[-300, -300]$. In this environment, GPI-LS and PCN are struggling to sufficiently explore and learn meaningful policies along the velocity objective (i.e., the walkers learn mostly to stand still), in comparison to the results from Figure 5. This highlights a disadvantage of centralisation, especially in terms of the increase in action space (i.e., a composition of each walker's action space—a vector of size 4, with values between -1 and 1).

We note that while MOMAPPO, PCN, and GPI-LS have all been evaluated on *multi_walker*, we do not present their results in a single unified plot, as they each use a different pipeline for arriving at the solution set. Namely, MOMAPPO repeatedly transforms the setting into a MARL problem using different linear scalarisation functions, and then applies MAPPO on each instance, while for PCN and GPI-LS, we first translate the setting to a MORL problem

via the *CentraliseAgent* wrapper. As a result, the approaches are also trained on different observation/action spaces (i.e., MOMAPPO supports decentralized execution, whereas the centralised single-agent MORL baselines do not), and for different number of iterations (e.g., given the size of the observation and action spaces in the centralisation case, it was computationally infeasible to train GPI-LS for the same amount of iterations as MOMAPPO (see Appendix B.1 for compute times). Since these algorithms address fundamentally different problem settings, we chose to present them separately.

## 6.2 Individual Reward and Known Utility Function

Here we consider the setting of independent learners, and known linear utility functions, reducing the problem to independent multi-agent RL. To demonstrate this setup, we run experiments on two MOMALAND environments, namely *mobeach_v0* and *moroute_choice_v0*. The considered learning approach is scalarised independent Q-learning (IQL) (Mannion et al., 2018), since we investigate congestion domains that are fairly simple in terms of state and actions spaces, but that involve a large number of agents (i.e., 50, 100 and 4200 agents).

**MO-BPD**  For evaluating the *Multi-Objective Beach Problem Domain*, we aim to reproduce the empirical studies performed by Mannion et al. (2018). We note that in comparison to the original MO-BPD, evaluated under tabular approaches, MOMALAND augments the agents' individual observation with additional information[6], potentially requiring function approximation-based techniques. However, MOMALAND also includes an additional wrapper, to make the environment equivalent to the original one introduced in Mannion et al. (2018). For *mobeach_v0*, using the aforementioned wrapper, our setup is identical to the two empirical studies of Mannion et al. (2018). First we consider 50 agents (35 type A, 15 type B), with 5 sections, each with capacity 3. The second setup includes 100 agents (70 type A, 30 type B), 5 sections, all with capacity 5. In both cases, agents have homogeneous preferences over the two objectives, with weights equal to $[0.5, 0.5]$. We also use a fixed initial distribution of agents over sections (half of the agents start in section 1 and half in section 3). Additional experimental details are available in Appendix B.

Figure 9 presents the learning curves of scalarised IQL. The independent Q-learners are studied under two different vectorial reward signals, namely individual reward (each agent receives the reward corresponding to its local section) and team reward (agents receive the same reward describing the entire beach, but still use an individual learning approach). Our results match the ones presented in Mannion et al. (2018): the team reward represents a more informative signal for the independent learning setup, leading to better performance. We also notice that the initial exploration, at the start of the learning process, leads agents to a higher averaged scalarised reward. However, the independent Q-learners are not able to retain the configuration, signalling that additional coordination mechanisms are required for this setting.

**MO-RouteChoice**  For *moroute_choice_v0*, we consider the Braess's paradox (Braess, 1968) road network, depicted in Figure 10(a), with 4200 agents that can choose between three routes (i.e, (1) $s - v - t$, (2) $s - w - t$, (3) $s - v - w - t$), to travel from the starting node $s$, to the destination node $t$. The reward for the travel time component is depicted on

---

[6]https://momaland.farama.org/environments/mobeach/

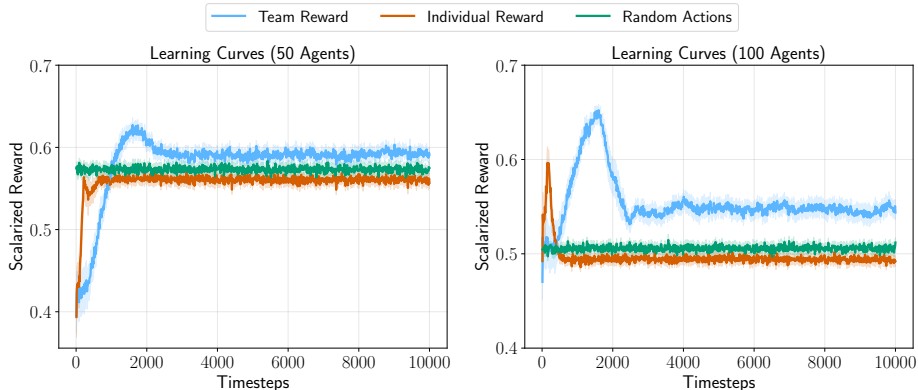

FIGURE 9: Average and 95% confidence intervals of scalarised team reward on training results from IQL on *mobeach_v0*.

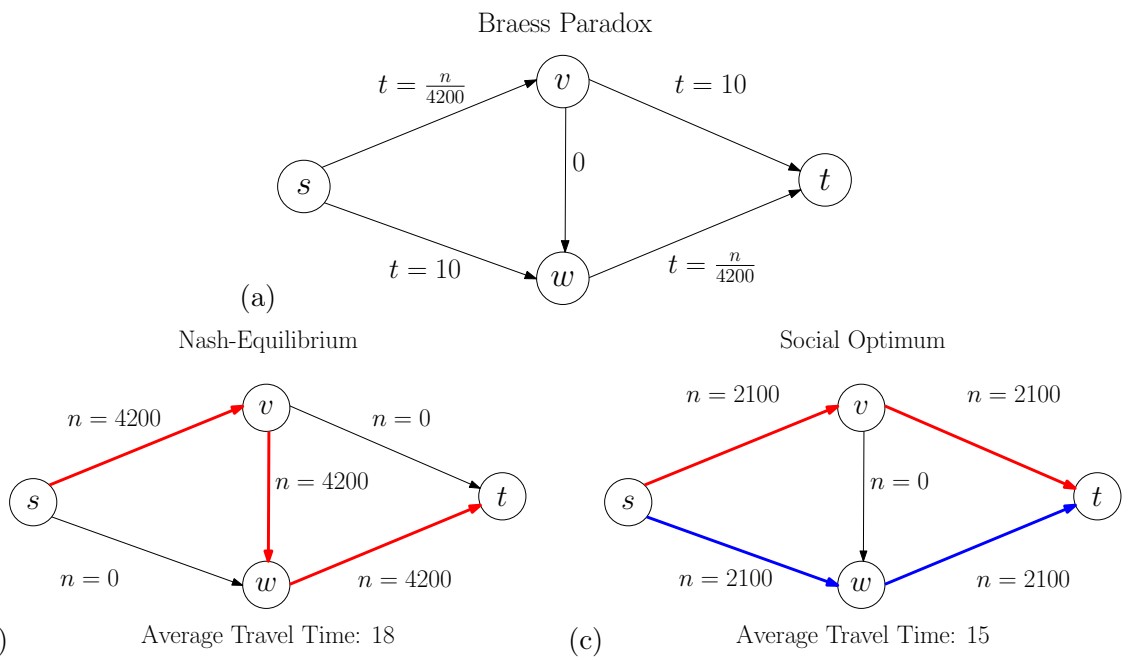

FIGURE 10: Route network for *moroute_choice_v0* and the corresponding reward for the time component for each edge, together with the configuration of the Nash equilibrium and social optimum for this setting.

each edge of the road network, while the cost component is calculated using the marginal cost tolling scheme (more details on the reward function are presented in Appendix C).

Congestion problems exhibiting the Braess's paradox, under the travel time reward component, have already been studied in the literature Wolpert and Tumer (2002); Valiant and Roughgarden (2006); Rădulescu et al. (2017). This class of problems famously demonstrates the effect of the tragedy of the commons, in which the selfish maximisation of resource use at an individual level will lead to worse outcomes at a societal level. We illustrate this

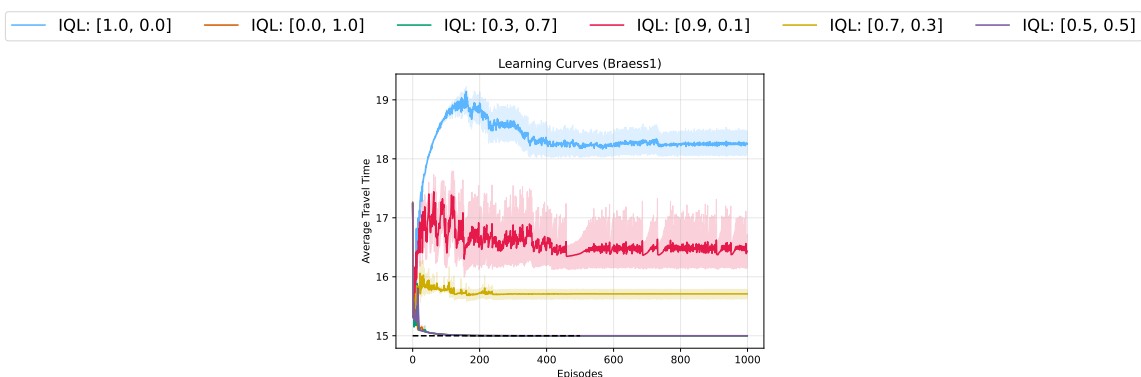

FIGURE 11: Average and 95% confidence intervals of travel time on training results from IQL on *moroute_choice_v0.*

phenomenon in Figure 10(b) versus Figure 10(c): the Nash equilibrium leads all agents to select route (3) $s - v - w - t$ for an average travel time of 18, while the social optimum of the system is for agents to ignore section $v - w$, and to equally split between the remaining two routes, for an average travel time of 15.

In Figure 11 we analyse the behaviour of the independent Q-learners, under different linear utility functions. For example `IQL` : $[0.3, 0.7]$ denotes the setting in which all agents assign a weight of 0.3 for the travel time objective and 0.7 for the cost objective. When all agents exclusively value the travel time objective (i.e., $[1.0, 0.0]$), the population converges to the Nash equilibrium (Figure 10b), with the worst outcome for the average travel time, 18. This phenomenon is mitigated when even a small mass is shifted towards the cost objective (e.g., $[0.9, 0.1]$). We notice how agents converge to the social optimum in the cases in which the weight for the cost objective is $\geq 0.5$, with an average travel time of 15. These results are in line with the work of Ramos et al. (2020a), that demonstrated that marginal-cost tolling leads agents to socially-desirable outcomes.

## 7 Open Challenges

In this section, we highlight some of the key challenges for future research on MOMARL.

### 7.1 Solution Concepts for MOMARL

In Section 3.2 we briefly outlined some possible solution concepts for MOMARL, focusing on the two main approaches in the literature: the axiomatic approach and the utility-based approach. To date, the utility-based approach has generally been the most common approach for MOMARL problems, as it allows for prior knowledge about the agents' preferences over objectives to be incorporated to simplify the problem.

When following the utility-based approach, solution concepts from traditional single-objective game theory can be extended to multi-objective settings by measuring agent incentives with respect to individual utility (rather than with respect to individual rewards/-payoffs in single-objective game theory). For example, Rădulescu et al. (2020b) extended the well-known Nash equilibrium and correlated equilibrium solution concepts to MOMA settings

using the utility-based perspective. Much of the analysis to date on solution concepts has focused on stateless single-shot settings (MONFGs), so further empirical studies are required in sequential settings. Extending existing solution concepts to MOMA settings is not trivial when following the utility-based approach, as one must consider the effect of the choice of optimisation criterion, either SER or ESR, as outlined in Section 3.2. The choice of the correct optimisation criterion is crucial when the utility functions are non-linear; selecting SER in place of ESR (or vice versa) can drastically alter the collective behaviour of the agents. For example, it has been demonstrated that it may not be possible for agents to reach a stable outcome, e.g., Nash equilibria may not exist under SER (Rădulescu et al., 2020a) or stable coalitions may not exist in coalition formation games (Igarashi and Roijers, 2017). It is also possible to have a mixture of optimisation criteria within the same system, where some agents follow SER and others follow ESR (Röpke et al., 2022). Work on such settings has been extremely limited to date and therefore further work is required to better understand the implications of mixed optimisation criteria.

Research on the axiomatic approach to MOMA problems is even less mature than the utility-based approach. The axiomatic approach may be a suitable fallback in settings where no information is available about the agents' utilities, although the space of joint policies that could be optimal is potentially much larger when no information is available about the utilities. As shown in Section 6.1, applying the axiomatic approach in team reward settings, where all agents receive the same reward vectors, is relatively straightforward and the problem is fully cooperative as all agent incentives are perfectly aligned. The Pareto optimal set in team reward settings simply includes all joint policies where the return vector is non-dominated. For individual reward settings (e.g., adversarial or mixed settings), Pareto optimal sets could be defined individually for each agent, as a joint policy that is Pareto optimal with respect to one agent's reward function may not necessarily be Pareto optimal for other agents. Such individual Pareto optimal sets would need to be conditioned on the behaviour of other agents in the system, so would in effect be a set of non-dominated responses to the other agents' policies (Rădulescu et al., 2020a). When policies are deterministic with a finite number of discrete actions, the non-dominated response set for an agent would also have a finite number of policies. In settings with probabilistic policies, the non-dominated response set could potentially have an infinite number of policies.

Finally, the relationship between the axiomatic and utility-based approaches in MOMA systems is currently not well understood and merits further study. Initial work by Mannion and Rădulescu (2023) in a team reward individual utility setting demonstrated that it is possible to have settings where none of the Nash equilibria are Pareto optimal, depending on the preferences of agents over objectives.

## 7.2 Utility Modelling and Preference Elicitation

In single-agent settings, it is possible to elicit and align preferences with respect to different trade-offs between objectives by directly interacting with the users (Peschl et al., 2022; Roijers et al., 2017). This is because it is beneficial for both the agent and the user to share such preferences openly. In multi-agent team utility settings, this would still be the case.

However, once we find ourselves in the individual utility case, the process becomes significantly harder. One may look at the problem from multiple perspectives: agents can

interact and model the preferences of their users, however agents can now also potentially model their opponents' utility function, in order to gain an advantage in the strategic interactions. To the best of our knowledge, interactive MOMARL, where agents have to concurrently learn their associated user's preferences, as well as how to optimally act in the environment, has not yet been explored. Overcoming the difficulties posed by misalignment of preferences, as well as the fact that it might no longer be in the agents' best interest to share their preferences openly (on the contrary, it might even be better to actively hide this information) are still very much open challenges. Potential directions for approaching these challenges include negotiation (Filipczuk et al., 2022; Baarslag, 2024), or social contracts (Hédoin, 2021).

### 7.3 Algorithms and Environments for MOMARL

Because it is a relatively new area, limited research has been focused on MOMARL. Moreover, although there is a wealth of problems documented in the literature that involve both multiple agents and objectives (Wurman et al., 2022; Pepper and Thomas, 2024), they are often simplified and not treated as MOMA. This prevents easy identification of contributions and comparison to the current state of the art in the field.

Consequently, few solving methods addressing both dimensions of the problem exist. Indeed, most works operate in the known utility setting, effectively relying on or adapting MARL methods, e.g. Mannion et al. (2018); Rădulescu et al. (2021). A notable exception to this is MO-MIX (Hu et al., 2023), which is able to learn a Pareto set of multi-agent policies in the team reward setting. As previously stated, additional research is required in general settings to establish solution concepts and develop algorithms that can identify these.

Before MOMAland, very few environments have been identified, modelled, and made available as MOMA problems. Although we offer a preliminary set of intriguing challenges, we think this collection can be expanded and invite external contributions of new and interesting environments. For instance, the majority of the suggested environments lack a known optimal Pareto front. Knowing the optimal Pareto front would enable algorithm developers to confirm the optimality of their approaches. Another example would be contributing MOBG or MOCBG environments to the library (Figure 2). We also invite collaborations and proposals of domains based on industrial applications, especially involving environments with stochastic dynamics. Finally, recent advances in frameworks like JAX (Bradbury et al., 2018) offer significant potential for acceleration through hardware-efficient computation. Although MOMAland does not yet support JAX-based environments, we consider this a promising direction for future development.

Hence, by making MOMAland open-source and open to contributions, we hope to receive external contributions of new algorithms and environments from the research community.

### 8 Conclusion

In this paper, we introduced MOMAland, the first publicly available benchmark suite for MOMARL problems. Our library includes a collection of over 10 environments under two different APIs for turn-based or simultaneous actions. These environments offer a diverse set of challenges, varying in the number of agents, state and action spaces, reward structures, and utility considerations. Notably, some of these challenges have no known solution concept.

We showed how to leverage existing literature from both multi-objective RL and multi-agent RL to construct new MOMARL algorithms able to solve some of the presented challenges. These baselines, along with useful utilities, are also made available to help algorithm designers in their future research endeavours.

While the release of MOMAland addresses one of the key challenges required to progress the field of MOMARL, many open challenges remain, as highlighted in Section 7. We hope this benchmark suite will be a valuable asset to the research community and that our work will inspire and enable future progress in the field.

## 9 Broader Impact Statement

Many real-world applications are inherently multi-objective and multi-agent, yet current RL methods fall short of addressing these challenges effectively. Developing methods within a framework that explicitly encompasses these dimensions will allow for richer solution sets that explore the trade-offs among objectives, and the dependencies between agents. We believe that decision-support systems stemming from this foundation will enhance users' agency, exhibit better alignment capabilities, and higher potential to support characteristics such as explainability, fairness, or transparency, in contrast to current systems that take a monolithic perspective and learn behaviors based on maximizing a single feedback signal. In summary, we believe MOMARL represents a vital extension for successfully deploying RL in real-world scenarios.

This article and MOMAland mark a pivotal step in formalizing and advancing research in this critical field. The introduced environments are designed to cover a diverse set of challenges, in terms of number of agents, problem dimension, reward and utility structure. Caution must be taken when developing MOMARL algorithms and ensure the evaluation is carried out on a diverse set of environments, to avoid overfitting to certain problem, reward or utility classes. Furthermore, the current benchmarks are limited to toy scenarios. While this can stimulate and support initial progress in the field, as future work, we aspire to include real-world scenarios that can further drive algorithmic development.

## 10 Affiliations

[1] SnT, University of Luxembourg, Luxembourg
[2] Farama Foundation, USA
[3] ETH Zurich, Switzerland
[4] AI lab, Vrije Universiteit Brussel, Belgium
[5] Centrum Wiskunde & Informatica, the Netherlands
[6] Information Systems group, Eindhoven University of Technology, the Netherlands
[7] School of Computer Science, University of Galway, Ireland
[8] Innovation, DII, City of Amsterdam, the Netherlands
[9] FSTM/DCS, University of Luxembourg, Luxembourg
[10] CNRS/CRIStAL, University of Lille, France
[11] Intelligent Systems group, Utrecht University, the Netherlands

**Acknowledgments and Disclosure of Funding**

This work was supported by the Fonds National de la Recherche Luxembourg (FNR), CORE program under the ADARS Project, ref. C20/IS/14762457, and by funding from the Flemish Government under the "Onderzoeksprogramma Artificiële Intelligentie (AI) Vlaanderen" program, and by the FWO, grant number G062819N. It has also received funding from the project ALIGN4Energy (NWA.1389.20.251) of the research programme NWA ORC 2020 of the Dutch Research Council (NWO), and from the European Union's Horizon Europe Research and Innovation Programme under Grant Agreement number 101120406. The paper reflects only the authors' view and the EC is not responsible for any use that may be made of the information it contains. Roxana Rădulescu was partly supported by the Research Foundation Flanders (FWO), grant number 1286223N. Willem Röpke is supported by the Research Foundation – Flanders (FWO), grant number 1197622N. Hicham Azmani is supported by the Research Foundation – Flanders (FWO), grant number 1SA9826N. We would also like to thank Lucas N. Alegre for his valuable inputs, and Manuel Goulao for helping us with the website.

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

## Appendix A. General Information

### A.1 Links

- Our documentation is accessible at: `https://momaland.farama.org/`;

- Our code is accessible at `https://github.com/Farama-Foundation/momaland`;

- Tracked experiments: `https://wandb.ai/openrlbenchmark/MOMAland`, `https://wandb.ai/rradules/MOMAland-IG-3`, and `https://wandb.ai/rradules/MOMAland-MW`

### A.2 Licenses

MOMAland is distributed under the GPL-3.0 license. We, the authors, bear all responsibilities in case of violation of rights.

### A.3 Maintenance

MOMAland is maintained by the authors, with the support of the community and the Farama Foundation team.[7]

## Appendix B. Reproducibility

|  | Hyperparameter | Value |
|---|---|---|
| **MAPPO** | Actor hidden layers | [256, 256] |
|  | Critic hidden layers | [256, 256] |
|  | Activation | tanh |
|  | Anneal learning rate | true |
|  | Clip epsilon | 0.2 |
|  | Entropy coefficient | 0. |
|  | $\gamma$ | 0.99 |
|  | GAE lambda | 0.99 |
|  | Learning rate | 0.001 |
|  | Max grad norm | 0.5 |
|  | Number of minibatches | 2 |
|  | Number of steps per epoch | 1280 |
|  | Updates per epoch | 2 |
|  | VF coefficient | 0.8 |
| **MOMAPPO** | Timesteps per weight | 500,000 for *multiwalker*, 150,000 for *catch* |
|  | Num weights | 20 for *multiwalker*, 10 for *catch* |
|  | Weight generation | Uniform (Blank et al., 2021) |

TABLE 4: Hyperparameter values used for MOMAPPO.

---

[7] `https://farama.org/team`.

| | Hyperparameter | Value (IG) | Value (MW) |
|---|---|---|---|
| **GPI-LS** | Hidden layers | [256, 256] | [256, 256] |
| | Batch size | 256 | 128 |
| | $\gamma$ | 0.99 | 0.99 |
| | Initial $\epsilon$ | 1.0 | - |
| | Final $\epsilon$ | 0.05 | - |
| | $\epsilon$ decay steps | 75000 | - |
| | Prioritized experience replay | false | false |
| | Gradient updates | 10 | 3 |
| | Target net update frequency | 200 | 200 |
| | Learning rate | 0.0003 | 0.0003 |
| | Timesteps per iteration | 10,000 | 10,000 |
| | Policy noise | - | 0.05 |
| | Total timesteps | 150,000 | 5M |
| **PCN** | Learning rate | 0.001 | 0.001 |
| | Hidden layers | [256, 256] | [256, 256] |
| | Batch size | 256 | 256 |
| | $\gamma$ | 0.99 | 0.99 |
| | Scaling factor | 1 for all objs. and horizon | 1 for all objs. and horizon |
| | Max buffer size | 100 | 500 |
| | Num model updates | 50 | 100 |
| | Policy noise | - | 0.05 |
| | Total timesteps | 150,000 | 100M |

TABLE 5: Hyperparameter values used for GPI-LS and PCN.

Table 4 lists the hyperparameter values used to conduct the experiments involving MOMAPPO (Algorithm 1). Table 5 presents the hyperparameters used for the experiments involving the *CentraliseAgent* wrapper, for PCN and GPI-LS (Section 6.1.2). We report the hyperparameters that differ from the default values specified in the MORL-baselines repository (Felten et al., 2023). Table 6 reports the hyperparameters used for the IQL experiments on the *mobeach_v0* and *moroute_choice_v0* environments (Section 6.2).

| | Hyperparameter | Value |
|---|---|---|
| **IQL** | Learning Rate | 0.5 |
| | Learning Rate Decay | 1 |
| | Learning Rate Min. | 0. |
| | Exploration Rate | 0.05 |
| | Exploration Rate Decay | 0.9999 |
| | Exploration Rate Min. | 0. |
| | $\gamma$ | 0.9 |

TABLE 6: Hyperparameter values used for IQL.

## B.1 Computational Cost of Baselines

To provide practitioners with guidance on the computational requirements of different settings, we report approximate walltimes for representative experiments across environments and algorithms in Table 7. Our experiments were carried out on a mix of local hardware, the high-performance computing facilities of the University of Luxembourg (Varrette et al., 2014), and the Flemish Supercomputer Center (VSC) at Vrije Universiteit Brussel, funded by the Research Foundation—Flanders (FWO) and the Flemish Government. Tracked experiments are available in Open RL Benchmark (Huang et al., 2024), at `https://wandb.ai/rradules/MOMAland-IG-3`, and `https://wandb.ai/rradules/MOMAland-MW`.

TABLE 7: Approximate walltimes per run for selected algorithms and environments.

| Algorithm | Environment | Steps | Walltime per run |
|---|---|---|---|
| **MOMA algorithms** | | | |
| MOMAPPO | *mo_multiwalker* | 10M | ∼9h30 (UL HPC) |
| MOMAPPO | *catch* | 1.5M | ∼1h (MacBook Pro M3 Max) |
| **MORL algorithms** | | | |
| GPI-LS | *mo_multiwalker* | 5M | ∼5 days (VUB HPC) |
| PCN | *mo_multiwalker* | 100M | ∼1 day 4h (VUB HPC) |
| GPI-LS | *mo_item_gathering* | 150K | ∼12h (VUB HPC) |
| PCN | *mo_item_gathering* | 150K | ∼25min (VUB HPC) |
| **MA algorithms** | | | |
| IQL | *mo_route_choice* | 1000 | ∼25min (MacBook Pro M3) |
| IQL | *mo_beach* | 10000 | ∼5min (MacBook Pro M3) |

## Appendix C. Environment details

### C.1 Multi-Objective Beach Problem Domain (MO-BPD)

The Multi-Objective Beach Problem Domain (MO-BPD) was introduced by Mannion et al. (2018) and extends an earlier single-objective version introduced by Tumer and Proper (2013); Devlin et al. (2014). In MO-BPD, each agent represents a tourist starting at a specific beach section, and then deciding at which section of the beach they will spend their day. Agents can choose to move to an adjacent section (*move_left* or *move_right*), or to *stay_still*.

Each beach section is characterised by a capacity $\psi$ and each agent is characterised by one of two static types: $A$ or $B$. These properties, together with the location of the agents on the beach sections, determine the vectorial reward received by agents, having two conflicting objectives: "capacity" and "mixture".

The environment can be configured in two modes, "individual" or "team" reward: the agents can either receive their own individual local rewards, based on the beach section they are located in (i.e., individual reward setting), or the global reward, based on the sum of rewards over all the available beach sections (i.e., team reward setting).

The capacity reward function is designed to return the highest value when the number of agents present is equal to the capacity of the section. Sections which are either too crowded or too empty receive lower rewards. The local capacity reward $L_{cap}(b)$ for a particular section is calculated as:

$$L_{cap}(b) = x_b e^{\frac{-x_b}{\psi}} \tag{10}$$

where $b$ is the beach section, and $x_b$ is the number of agents present at that section. The global capacity reward is then defined as:

$$G_{cap} = \sum_{b \in \mathbf{B}} L_{cap}(b) \tag{11}$$

The maximum mixture reward for a section is received when the number of $A$ agents in attendance is equal to the number of $B$ agents, while sections with an unequal mixture of agents receive a lower reward as they are less desirable. The local mixture reward $L_{mix}(b)$ [8] for a particular section is calculated as:

$$L_{mix}(b) = \frac{\min(|A_b|, |B_b|)}{|A_b| + |B_b|} \tag{12}$$

where $|A_b|$ is the number of agents of type $A$ present at that section, $|B_b|$ is the number of agents of type $B$ present at that section. The global mixture utility can then be calculated as the summation of $L_{mix}(b)$ over all sections in MO-BPD:

$$G_{mix} = \sum_{b \in B} L_{mix}(b) \tag{13}$$

A drawback of the original version of the benchmark derived from the fact that rewards were specified per timestep, meaning that increasing the number of timesteps also changed the Pareto front. To rectify this issue in our implementation rewards are received only in the last timestep.

The congestion condition in MO-BPD is only available when the number of agents is greater than the total capacity of the sections. Furthermore, as noted in Mannion et al. (2018), there are a few additional ways to ensure that no trivial solutions exist: by using odd values for $\psi$ implies that $L_{cap}$ and $L_{mix}$ cannot both be maximised at the same time at any one section and by using different proportions of $A$ and $B$ agents.

In terms of mathematical frameworks, under the individual reward setting, the MO-BDP is a MOPOSG, while the team reward setting casts the problem as a MODec-POMDP.

## C.2 MO-ItemGathering

The MO-ItemGathering environment is an extension of the two agent problem from Källström and Heintz (2019). In the original setting, agents must collect red, green and yellow items (representing the objectives) in a 8x8 grid world. However, in their implementation, only

---

[8]We note that in the initial version of the environment by Mannion et al. (2018), the local mixture component was further normalised by the number of beach sections, diminishing the 'local' (i.e., individual) perspective of the signal.

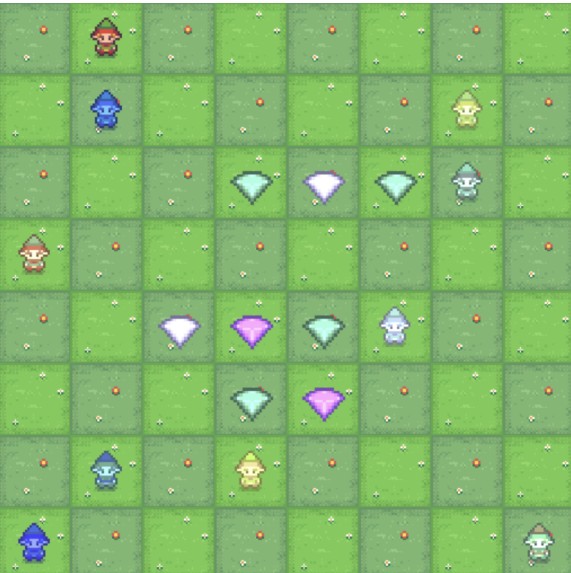

Figure 12: Illustration of the MO-ItemGathering with 10 agents and 3 objectives on a 8x8 grid.

one agent was controlled by MORL, the other agent used a hand-coded policy (always go for the closest red item). MOMAland extends the environment to any number of agents, objectives (i.e., item colours) and grid dimensions. An illustration of the environment is presented in Figure 12.

MO-ItemGathering is a fully observable environment, where the state received by an agent is a tuple comprising a matrix encoding the items' and agents' locations, together with the agent's id. The action space for each agent is discrete, with a size of 5, representing the cardinal directions and staying at the current position.

Agents acquire rewards upon stepping in a cell occupied by items, and receive a reward of +1 for the corresponding objective (i.e., colour). The vectorial reward has two modes, either an individual reward, where agents only receive rewards for the items they collect, or a team reward, where all agents receive a reward when an item is picked up in the environment. In the individual reward mode the environment is a MOSG, while in the team reward mode the environment is a MOMMPD.

### C.3 MO-GemMining

In Multi-Objective Gem Mining (which extends Gem Mining / Mining Day Bargiacchi et al. (2018); Roijers (2016) to multiple objectives), a mining company mines gems from a set of mines (local reward functions) located in the mountains (see Figure 13). The mine workers live in villages at the foot of the mountains. The company has one van in each village for transporting workers and must determine every morning to which mine each van should go.

Each village/van represents one agent. The action space of each agent is the set of mines that that agent can go. Please note that vans can only travel to nearby mines (which is represented in the graph connectivity). If multiple vans (from different villages) end up at the same mine, the total number of workers at those mines is summed. Workers are more

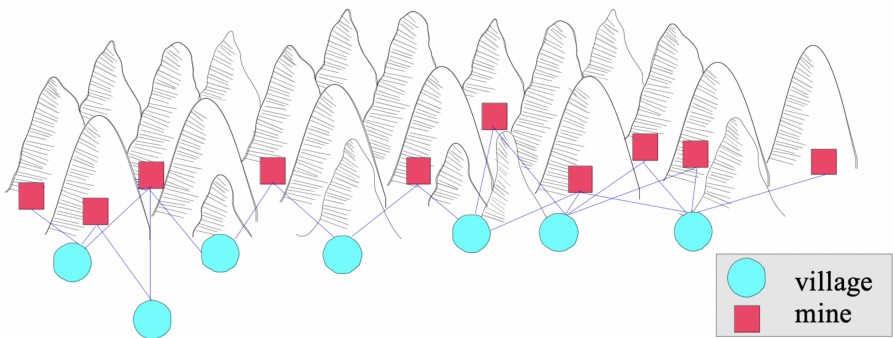

FIGURE 13: Illustration of the Multi-objective Gem Mining. Each village represents an agent, each mine represents a local reward function.

efficient when there are more workers at a mine: the probability of finding a gem of a given type in a mine is $\mathbf{x} \cdot b^{w-1}$, where $\mathbf{x}$ is the base probability of finding a gem of that type in the mine and $w$ is the number of workers at the mine. $b$ is a bonus factor per worker and has a $b = 1.03$ default. It is possible to truncate the probability of finding gems (across all types) to a maximum probability. By default, this truncation probability is set to 0.9. Please note that the truncation probability should not be higher than 1. When the number of workers at a mine is 0, no gems will be found. The number of gem types, i.e., objectives, is configurable.

We can generate instances of Multi-Objective Gem Mining for any number $v > 0$ of villages (agents). As a default, we randomly assign $1 - 5$ workers to each village and connect it to $2 - 4$ mines, but both lower and both upper limits are configurable. Each village is only connected to mines with a greater or equal index, i.e., if village $i$ is connected to $m$ mines, it is connected to mines $i$ to $i + m - 1$. The last village is connected to the maximum number of connected mines (default 4) and thus the number of mines is $v$ plus this maximum minus one (default $v + 3$).

The environment is a multi-objective multi-agent multi-armed bandit MOMAMAB, which extends the multi-agent multi-armed bandit (MOMAB) setting Bargiacchi et al. (2018); Verstraeten et al. (2020); Bargiacchi et al. (2022) to multiple objectives, and/or the multi-objective coordination graph (MOCoG) setting Roijers et al. (2013a, 2015) to reinforcement learning.

### C.4 Multi-Objective Route Choice Domain (MO-RouteChoice)

MO-RouteChoice is a multi-objective extension of the route choice problem Thomasini et al. (2023). In the route choice problem, $N$ independent drivers must choose routes to travel from a source to a destination while minimising their own travel time and considering the effect caused by other drivers.

The road network is represented as a directed graph $G = (V, E)$, where nodes $V$ correspond to intersections and links $E$ represent the roads/segments connecting them. Each link $l \in V$ has two associated costs: travel time ($c_l^T : x_l \to \mathbb{R}^+$) and monetary cost ($c_l^M : x_l \to \mathbb{R}^+$). Both costs are influenced by the flow $x_l$ of vehicles on the link. As more drivers use a specific link, it becomes more congested, increasing the costs for all drivers on that link.

During the initialisation of a chosen problem instance, each driver is assigned a fixed source node and destination node from the available origin-destination pairs. In each episode, every agent selects a route, which is a set of links connecting their assigned source and destination nodes. The costs associated with travelling a route are the sum of the costs of all its links: $C_R^T = \sum_{l \in R} c_l^T$ and $C_R^M = \sum_{l \in R} c_l^M$. The set of possible routes for each agent is predefined based on their assigned origin-destination pair and the available routes in the chosen problem.

MO-RouteChoice has two individual objectives: minimising travel time and minimising monetary cost. The monetary cost can be assigned to routes in two ways: either a random percentage of roads are tolled (controlled by a parameter), or all roads are tolled based on their occupancy (marginal-cost tolling).

The environment contains all road networks from the original route choice problem Ramos et al. (2020b); Thomasini et al. (2023). This includes Braess paradoxes and their extensions, as well as larger road networks inspired by real-world cities.

An initial MO perspective on this problem, with linear preferences, is presented in Ramos et al. (2020b). MO-RouteChoice is a MONFG.

## C.5 Multi-Objective PettingZoo Environments

This section describes PettingZoo Terry et al. (2021) environments that have been adapted to multi-objective settings. Documentation related to the original versions of these environments can be found at `https://pettingzoo.farama.org/`.

### C.5.1 MO-PistonBall

The MO-PistonBall environment is a MOPOSG that features multiple pistons whose overall goal is to move a ball to the edge of the screen. We keep all environment dynamics as in the original PettingZoo implementation, which ensures that the environment is partially observable as each piston can only observe its neighbours. Furthermore, while the environment is technically stochastic, the only stochasticity comes from selecting the initial start state. For our multi-objective extension, we decompose the original reward function, which was a linear combination of three components. Our reward function is defined as follows,

$$R_i^1(s_t, \boldsymbol{a}_t) = 100 \times \frac{x_t - x_{\text{ball}}}{x_0 - x_{\text{wall}}} \quad \text{(global reward)} \tag{14}$$

$$R_i^2(s_t, \boldsymbol{a}_t) = \frac{x_t - x_{\text{ball}}}{2} \quad \text{local reward for every piston under the ball} \tag{15}$$

$$R_i^3(s_t, \boldsymbol{a}_t) = -0.1 \quad \text{global time penalty unless the episode is over} \tag{16}$$

In this reward function, Equation (14) is a global reward shared by all agents and measures the distance travelled by the ball in the latest step, while Equation (15) is a local reward that is only received by pistons that are underneath the ball at that time. Finally, Equation (16) is a time penalty that all agents receive for every timestep that the ball has not reached the wall yet.

While the reward function was originally intended to induce a common goal, the individual nature of Equation (15) may result in unexpected results. For example, it may be possible

to push the ball, hoping that it bounces back such that it can obtain additional rewards. We note, however, that we have not yet observed such behaviour.

The environment is a multi-objective POSG where the only environment stochasticity comes from the initial state distribution.

### C.5.2 MO-MultiWalker-Stability (MO-MW-Stability)

This problem is a MODec-POMDP that involves multiple bipedal walkers (agents) collaboratively carrying a package to the right side of the screen without falling. The package is placed on top of the walkers at the beginning of each episode, is so large that it stretches across all walkers, and too large for any single walker to move on its own. In the original version of MultiWalker Gupta et al. (2017); Terry et al. (2021), each walker $i$ receives an individual reward defined by:

$$
R_i(s, \boldsymbol{a}) = \begin{cases} -100, & \text{package fallen, or package on the left} & (17) \\ -110, & \text{walker fallen and terminate on fall enabled} & (18) \\ \texttt{shaped} - 10, & \text{walker fallen and not terminate on fall} & (19) \\ \texttt{shaped}, & \text{otherwise.} & (20) \end{cases}
$$

Where:
$$
\texttt{shaped} = \frac{\texttt{forward\_reward} \times \Delta x_{\texttt{package}} \times 130}{\texttt{SCALE}} - 5 \times \Delta\texttt{angle}_{\texttt{head}},
$$

In Equations (17)–(20), line 17 penalizes the agents in case of package fall or going in the wrong direction, line 18 terminates the game with a penalty if one walker falls, lines 19 and 20 use a shaped reward to make the package move forward, as well as avoid brutal change of angle of the walker's head. Then, the rewards are combined as $r = \texttt{local\_ratio} \times r_i + (1 - \texttt{local\_ratio}) \times \overline{r_i}$, where $\overline{r_i}$ is the average individual reward.

Our multi-objective version of multi-walker adds another dimension to the problem, that is keeping the package stable by reducing the angle changes of the package. Essentially, the new reward dimension is identical to Equations (17)–(20), but replaces the `shaped` reward by $\texttt{stability} = \Delta\texttt{angle}_{\texttt{package}}$.

In this version, the rewards are defined as follows:

$$
R_i^1(s, \boldsymbol{a}) = \begin{cases} -100, & \text{package fallen, or package on the left} \\ -110, & \text{walker fallen and terminate on fall enabled} \\ \texttt{shaped} - 10, & \text{walker fallen and not terminate on fall} \\ \texttt{shaped}, & \text{otherwise.} \end{cases}
$$

$$
R_i^2(s, \boldsymbol{a}) = \begin{cases} -100, & \text{package fallen, or package on the left} \\ -110, & \text{walker fallen and terminate on fall enabled} \\ \texttt{stability} - 10, & \text{walker fallen and not terminate on fall} \\ \texttt{stability}, & \text{otherwise.} \end{cases}
$$

This environment is cooperative and agents only have a partial view of the global state. Hence, it is a MODec-POMDP.

## C.6 CrazyRL

The CrazyRL[9] environments are 3 MOMMDPs designed to facilitate the learning of high-level swarm formations around potentially moving objects for multiple drones Felten (2024). These environments rely on high-level control commands, such as a 3D speed vector, indicating where each drone should go, rather than low-level control like torque in the engine. This choice significantly simplifies the state and action spaces of the agents, enabling them to focus on the core problem of learning formation and eliminating the need for heavier robotics simulators such as Gazebo Koenig and Howard (2004) or Pybullet Coumans and Bai (2016); Tai et al. (2023).

In practice, each agent (drone) perceives its current $x$, $y$, and $z$ coordinates (denoted as $x_i$, $y_i$, $z_i$ for agent $i$) along with the target coordinates ($x_{\text{targ}}$, $y_{\text{targ}}$, $z_{\text{targ}}$). Additionally, agents also perceive the positions of other agents, making the environment fully observable. At each time step, agents select a 3D speed vector as their action, dictating the direction in which they wish to move, i.e. $a_i \in [-1, 1]^3$, $\forall i \in [1, n]$. The drones' movements are discrete, with their positions updated at each step by applying these action vectors, effectively "teleporting" them. If the moves lead outside coordinates specified by the map size, the new coordinates of the agents are clipped to stay inside the map. The global state is a concatenation of all known positions (agents and targets). Episodes terminate upon drone collisions, contact with the floor or target, or when a predefined number of time steps is reached.

In the three specified environments, the reward function encompasses two conflicting objectives: (1) minimizing the distance to the shared target while simultaneously (2) maximizing the distance from the other agents. In multi-objective settings, the goal is generally to maximize all objectives, requiring the need to transform the minimizing objective into a maximization of its negation. However, we noticed that transforming the first reward component into a negative value and maximizing both components can adversely affect learning performance on the studied policy optimization algorithm (PPO) Schulman et al. (2017). Therefore, we opted to convert the first objective into a potential-based reward instead Ng et al. (1999).

For each agent $i$, the rewards can be formally defined as follows:

$$R_i^1(s, \boldsymbol{a}) = \|(x_i^{t-1}, y_i^{t-1}, z_i^{t-1}) - (x_{\text{targ}}^{t-1}, y_{\text{targ}}^{t-1}, z_{\text{targ}}^{t-1})\|^2$$
$$- \|(x_i^t, y_i^t, z_i^t) - (x_{\text{targ}}^{t-1}, y_{\text{targ}}^{t-1}, z_{\text{targ}}^{t-1})\|^2,$$

$$R_i^2(s, \boldsymbol{a}) = \frac{\sum_{j \neq i} \|(x_i^t, y_i^t, z_i^t) - (x_j^t, y_j^t, z_j^t)\|^2}{n - 1},$$

where $R_i^o(s, \boldsymbol{a})$ is the $o^{\text{th}}$ objective value of agent $i$, and $x_i^t$ denotes the $x$ position of agent $i$ at time step $t$. These individual rewards are then aggregated to form a multi-objective team reward: $\boldsymbol{R}(s, \boldsymbol{a}) = \sum_{i \in [1, n]} \boldsymbol{R}_i(s, \boldsymbol{a})$.

These environments are cooperative and agents can perceive the location of everyone else. Hence, they are all MOMMDPs.

---

[9]An implementation of these environments in Jax (Bradbury et al., 2018) (Figure 3) and the code to fly real drones are available in the original repository: `https://github.com/ffelten/CrazyRL`. The name "CrazyRL" is a contraction of Crazyflie drones (Giernacki et al., 2017) and RL.

**Surround**    In this environment, the objective is for the drones to establish a stable formation around a fixed target.

**Escort**    This environment is an extension of the previous one and introduces the added challenge of a moving target. In this scenario, the target is assigned an initial position and a final position, and it moves linearly from the former to the latter in a specified number of time steps.

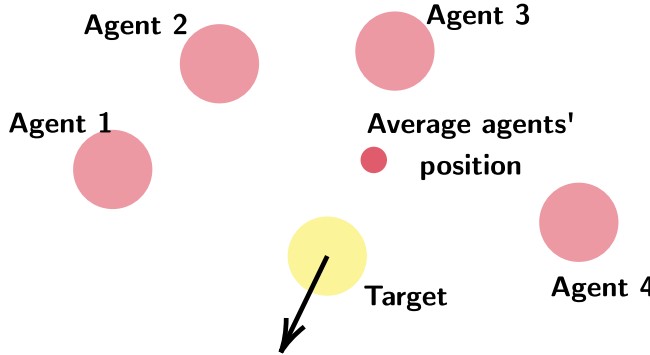

FIGURE 14: Target move decision in the Catch environment (flattened to 2 dimensions for illustrative purpose).

**Catch**    In this last environment, an element of intelligence is introduced into the target's behaviour. Specifically, the target tries to escape the agents by calculating an average position of these and endeavours to move in the opposite direction. However, if the computed average position is too close to the current target position, the target resorts to random movement as a strategy. An example of this strategy in 2 dimensions is exposed in Figure 14.

## C.7 Multi-Objective Board Games

### C.7.1 MO-Breakthrough

MO-Breakthrough is a multi-objective variant of the two-player, single-objective turn-based board game Breakthrough. In Breakthrough, players start with two rows of identical pieces in front of them and try to reach the opponent's home row with any piece. The first player to move a piece on their opponent's home row wins. Players move alternatingly, and each piece can move one square straight forward or diagonally forward. Opponent pieces can also be captured, but only by moving diagonally forward, not straight.

MO-Breakthrough optionally extends this game with one to three additional objectives: a second objective that incentivizes faster wins, a third one for capturing opponent pieces, and a fourth one for avoiding the capture of the agent's own pieces. The various possible trade-offs between these objectives could lead to e.g. more aggressive vs. more defensive playstyles, or more patient strategies aiming at winning eventually vs. more risky strategies aiming at winning quickly. Additionally, the board width can be modified from 3 to 20 squares, and the board height from 5 to 20 squares.

As the game is competitive and fully observable, MO-Breakthrough falls into the category of MOSGs.

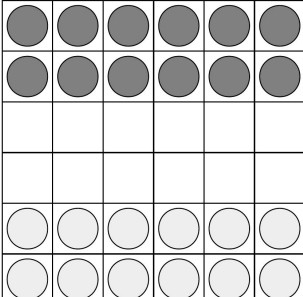 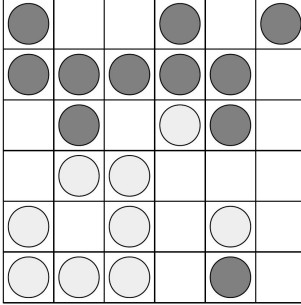

FIGURE 15: (MO-)Breakthrough (adapted from Baier, 2015). The left-side image shows the starting position of the game, and the right side shows a possible terminal position in which Black won.

### C.7.2 MO-CONNECT4

MO-Connect4 is a multi-objective variant of the two-player, single-objective turn-based board game Connect 4. In Connect 4, players can win by connecting four of their tokens vertically, horizontally or diagonally. The players drop their respective tokens in a column of a standing board (of width 7 and height 6 by default), where each token will fall until it reaches the bottom of the column or lands on top of an existing token. Players cannot place a token in a full column, and the game ends when either a player has made a sequence of 4 tokens, or when all columns have been filled (draw).

MO-Connect4 extends this game with a second objective that incentivizes faster wins, and optionally one additional objective per column for having more tokens than the opponent in that column. While the default objective of winning and the second objective of winning quickly could allow for finding trade-offs between more patient and more risky playstyles, the column objectives are more directly in conflict with each other, as having more tokens in one column means having fewer in another. Different trade-offs between them will lead to strategies that e.g. favour one side of the board over another, and are relatively easy to validate. Additionally, the width and height of the board can be set to values from 4 to 20.

MO-Connect4 is competitive and fully observable and therefore a MOSG.

### C.7.3 MO-INGENIOUS

MO-Ingenious is inspired by a competitive, turn-based board game for multiple players (BoardGameGeek, 2004). 2-6 players can play (default is 2), on a hexagonal board with an edge length of 3-10 (default is number of players + 4). Each player has 2-6 (default is 6) tiles with colour symbols on their rack, which is only observable to themselves (in the default rules). In sequential order, players play one of their tiles onto the hexagonal board, with the goal of establishing lines of matching symbols emerging from the placed tile. This allows the players to increase their score in the respective colours, each colour representing one of 2-6 (default is 6) objectives. After a tile has been played, a new one is randomly drawn into the player's rack.

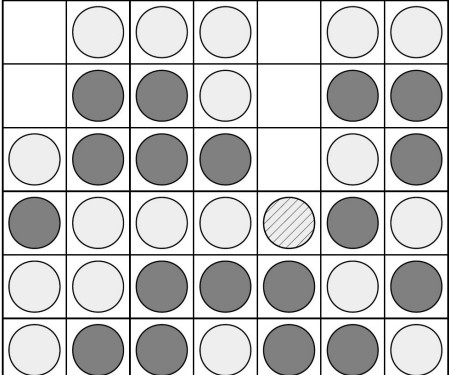

FIGURE 16: (MO-)Connect4 (adapted from Baier, 2015). White won the game by playing the marked move.

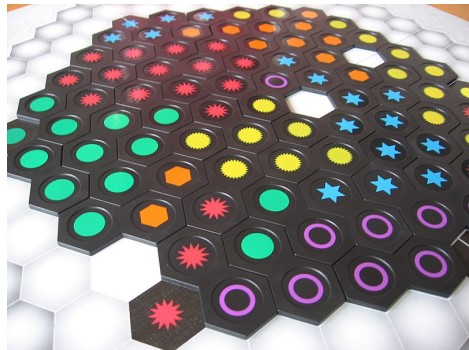

FIGURE 17: Ingenious (public domain image retrieved from Wikimedia, 2007).

When the board is filled, the original game rules define the winner as the player who has the highest score in their lowest-scoring colour; for a player with (red=5, green=2, blue=9) for example, the relevant score would be 2. Our implementation exposes the colour scores themselves as different objectives, allowing arbitrary utility functions to be defined over them by the user. In addition, MO-Ingenious extends the original game rules with an optional *team reward* mode, in which agents share all scores and play cooperatively, and an optional *fully observable* mode, in which agents can observe all racks.

In terms of mathematical frameworks, this environment is therefore a MOPOSG, which can be configured to become a MODec-POMDP when playing in team reward mode, a MOSG when playing in fully observable mode, or a MOMMDP when using both.

## C.8  MO-SameGame

MO-SameGame is a multi-objective, multi-agent variant of the single-player, single-objective turn-based puzzle game called SameGame. 1 to 5 agents can play (default is 1), on a rectangular board with width and height from 3 to 30 squares (defaults are 15), which are initially filled with randomly coloured tiles in 2 to 10 different colours (default is 5). Players move alternatingly by selecting any tile in a group of at least 2 vertically and/or horizontally

connected tiles of the same colour. This group then disappears from the board. Tiles that were above the removed group "fall down" to close any vertical gaps; when entire columns of tiles become empty, all columns to the right move left to close the horizontal gap.

The original single-player, single-objective SameGame rewards the player with $n^2$ points for removing any group of n tiles. MO-SameGame can extend this in two ways. Agents can either only get points for their own actions, leading to competition between them, or all rewards can be shared in "team reward" mode. Additionally, points for every colour can be counted as separate objectives, allowing for different trade-offs between colours, or they can be accumulated in a single objective like in the default game variant, essentially providing a single-objective wrapper for the game.

MO-SameGame is fully observable and can therefore be modelled as a MOSG in individual reward mode, or a MOMMDP when using team rewards.

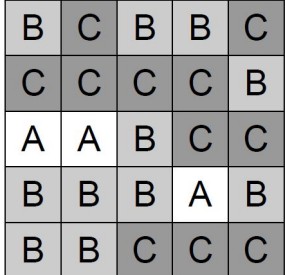

(a) A random start position on a 5×5 board with 3 colors.

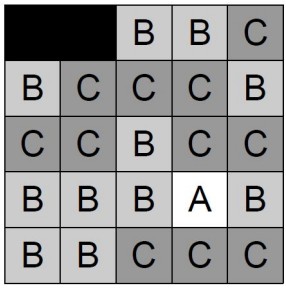

(b) Result after playing A in the leftmost column as first move.

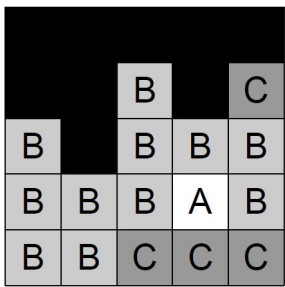

(c) Result after playing C in the leftmost column as second move.

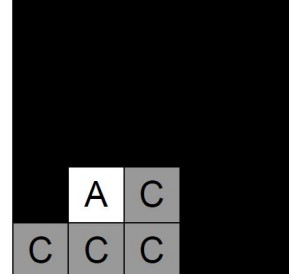

(d) Result after playing B in the leftmost column as third move.

FIGURE 18: The mechanics of SameGame (adapted from Baier, 2015).

