# OpenReview forum: "MOMAland: A Set of Benchmarks for Multi-Objective Multi-Agent Reinforcement Learning"
_DMLR — Accepted by DMLR_

### Review · Reviewer_or8V · 2025-06-01

**Recommendation:** 4
**Confidence:** 2

**Summary Of Contributions:**

This work introduces MOMAland, a gymnasium-like library for multi-objective multi-agent reinforcement learning. MOMAland consists of multiple environments spanning the axes of observability, cooperativeness, and statefulness. This work also evaluates standard algorithms on their environments, including translating standard MARL algorithms to MOMARL with scalarization wrappers and MORL algorithms with centralization wrappers.

**Strengths:**

I already listed the main strengths above. Overall, I feel this paper is well written and comprehensively introduces the nacent field of MOMARL. I think this work will greatly help the adoption of the MOMARL field by providing standardized environments and baseline algorithms.

**Audience:**

Yes

**Broader Impact Concerns:**

The Broader Impact statement is satisfactory.

**Claims And Evidence:**

Yes. Specifically, I have no concerns about the experiments in section 6.

**Datasets And Benchmarks:**

There is sufficient detail for this benchmark, because it is fully open source and hyperparameters are shared in the appendix.

**Extended Submissions:**

N/A

**Limitations:**

I think the current set of limitations presented by the authors in section 7 is satisfactory.

**Requested Changes:**

Clarifications:
- For figure 5, does the "timesteps axis" treat all policies as being generated in parallel? (I.e. at 5M timesteps, instead of 10 policies being fully trained it represents all 20 policies halfway trained) If so, I think this should be clarified in the text as it is slightly different from the sequential algorithm presented in algorithm 1.
- As listed in the weaknesses above, I would've liked to see a short explanation about why a researcher should choose a specific environment. For instance, the description of MO-RouteChoice on page 14 does not explain why this is a challenging learning problem whereas the deeper inspection on IQL on this environment in page 21-22 demosntrates why this is a social dilemma.

Additional experiments to strengthen the paper:
- I would've liked to see an experiment using nonlinear utility functions, just to show how the distinction between SER and ESR impacts the design of algorithms.

Typos:
- Page 7, after definition 5, it says the definition does not conflict with "theorem" 3, but I think it meant "definition" 3
- Page 9, there is a stray single quote in section 3.3 after "used"
- Fix the broken links in the appendix. There are a few "??" that refer to sections in the main paper and should be corrected.

**Strengths And Weaknesses:**

Strengths:
- As someone coming from a MARL background who was only aware of the MORL literature from a high level, I think this paper provides a clear introduction to the MOMARL, especially in terms of the detailed descriptions of solution concepts.
- This work addresses a major roadblock to MOMARL progress and adoption: the lack of standardized environments and interfaces. Furthermore, by integrating with the rest of the Farama ecosystem, MOMAland makes it easy to test MARL/MORL algorithms in MOMARL settings.
- The experiments in section 6 demonstrate how the library is able to reproduce the results of earlier works along with showing how MOMARL has a different set of challenges from single-objective MARL.

Weaknesses:
- Some of my issues with this work are inherited from the upstream projects that it follows. In particular, the PettingZoo interface (which MOMAland largely inherits) does not have a first-party vectorization API to the best of my knowledge (i.e. running multiple environments in parallel), making it inconvenient for researchers  who want faster training or custom environments that can run on the GPU (like jaxmarl among others).
- Despite comprehensively summarizing and categorizing each environment in MOMAland in section 5, it is still unclear to me what the purpose of each environment is. I would like a few sentences explaining why a researcher should choose a specific environment over others, like saying a setting has a significant exploration challenge or there is a social dilemma.
- All of the experiments use linear utility functions, making the distinction between SER, ESR, and immediately scalarizing rewards irrelevant. I would have appreciated an experiment with nonlinear utilities to demonstrate how it impacts the design of algorithms.

---

### Review · Reviewer_1q9W · 2025-06-11

**Recommendation:** 4
**Confidence:** 2

**Summary Of Contributions:**

This paper introduced a comprehensive benchmark suite MOMAland for MOMARL problems under over 10 environments. It discussed the detailed APIs and utilities in MOMAland, as well as several learning algorithms for addressing the involved environments. Thus, MOMAland will significantly advance the research and applications of MOMARL in the future.

**Strengths:**

- A comprehensive benchmark suite MOMAland for MOMARL is introduced in the paper.
- The environments, APIs and utilities in MOMAland are well explained.
- Open questions in MOMARL are discussed.

**Audience:**

Yes

**Claims And Evidence:**

The claims are well explained in the paper.

**Datasets And Benchmarks:**

The benchmarks are well motivated and explained.

**Extended Submissions:**

NA

**Limitations:**

One major concern is the evaluation of existing approaches, including the baselines implemented in Table 2, on the environments in the proposed benchmarks. More quantitative results can be provided to illustrate the performance of existing techniques, and provide more insights into understanding the benefits and weaknesses of previous method.

**Requested Changes:**

- More preliminary experimental results can be provided using the proposed benchmarks, e.g., the performance comparison of the baseline algorithms in Table 2 on various environments.

**Strengths And Weaknesses:**

Strengths:
- A comprehensive benchmark suite MOMAland for MOMARL is introduced in the paper.
- The environments, APIs and utilities in MOMAland are well explained.
- Open questions in MOMARL are discussed.

Weaknesses:
- More quantitative results can be provided to illustrate the performance of existing techniques, and provide more insights into understanding the benefits and weaknesses of previous method.

---

### Review · Reviewer_Jfo9 · 2025-09-07

**Recommendation:** 4
**Confidence:** 2

**Summary Of Contributions:**

The primary contribution of the paper is the introduction of MOMAland, the first standardized benchmark suite explicitly designed for multi-objective multi-agent reinforcement learning (MOMARL). While benchmarks such as Gymnasium, PettingZoo, and MO-Gymnasium have shaped progress in single-agent or single-objective domains, no prior framework supported the simultaneous presence of multiple agents and multiple objectives. MOMAland directly addresses this gap by providing over ten diverse environments, ranging from cooperative to competitive and mixed settings, with both discrete and continuous action/state spaces and varying observability conditions. This makes it a much-needed foundation for advancing research in an emerging but underdeveloped subfield of reinforcement learning.

Beyond introducing environments, the paper extends the APIs and utilities that facilitate modern RL experimentation. Building on the PettingZoo interface, MOMAland adapts the API to return vectorial rewards, thereby naturally capturing multi-objective settings. It offers both parallel and agent-environment cycle (AEC) APIs, ensuring usability across simultaneous and turn-based tasks. A collection of wrappers further enhances flexibility: rewards can be normalized, scalarized into single-objective signals for compatibility testing, or centralised to collapse the multi-agent dimension into a single learning agent. These utilities align MOMAland with the Farama Foundation ecosystem, ensuring seamless interoperability with popular tools such as Gymnasium, SuperSuit, and MORL-Baselines.

In addition to environments and APIs, the benchmark includes a set of baseline algorithms designed to provide initial reference points for the community. These baselines—such as MOMAPPO, scalarised Independent Q-Learning (IQL), Pareto Conditioned Networks (PCN), and Generalised Policy Improvement Linear Support (GPI-LS)—illustrate how existing MORL and MARL techniques can be adapted to MOMARL. While not intended to be state-of-the-art performers, these implementations offer a reproducible starting point and demonstrate the mechanics of leveraging MOMAland’s wrappers and evaluation tools. Crucially, the paper advocates for standardized evaluation using performance indicators well-established in the multi-objective optimization literature, including hypervolume, cardinality, and expected utility, thereby grounding MOMARL experimentation in principled metrics.

Finally, the authors provide empirical demonstrations and insights by running these baseline algorithms across several environments. The experiments reveal meaningful patterns, such as the trade-offs between formation compactness and collision avoidance in drone coordination tasks, or the emergence of inefficient equilibria in route-choice scenarios akin to Braess’s paradox. Results in the multi-objective beach domain show the difference between team-based and individual reward signals in independent learning, highlighting coordination challenges that persist in MOMARL. Through these examples, the paper not only validates the functionality of MOMAland but also surfaces research questions about the adequacy of existing algorithms, the importance of coordination signals, and the complexity of balancing multiple objectives in multi-agent contexts.

**Strengths:**

The full strength list can be found in my previous box. Here I only delve into the key contributions of the submission, detailed as below.

* The submission makes a significant contribution by introducing MOMAland, the first benchmark suite for multi-objective multi-agent reinforcement learning, filling a clear gap left by prior frameworks such as Gymnasium, PettingZoo, and MO-Gymnasium. This is both novel and timely, with strong potential impact on research in domains like traffic, energy, and resource allocation.

*  The work is well-grounded in prior literature, positioning itself as a natural extension of existing RL benchmarks while addressing the unique challenges of vectorial rewards and interacting agents. Its relevance to the community is high, as the open-source release and Farama integration ensure broad accessibility and adoption.

* In terms of quality and clarity, the paper provides rigorous formal definitions, principled evaluation metrics, and baseline implementations that, while simple, establish reproducible reference points. The presentation is clear, with useful figures and a systematic structure.

* Finally, the authors recognize the ethical and social implications of MOMARL, emphasizing that many real-world decision-making problems are inherently multi-objective and multi-agent. By supporting research in this space, the benchmark fosters progress toward more fair, transparent, and aligned AI systems.

**Audience:**

Yes

**Claims And Evidence:**

Overall, the claims in the submission are well supported and clearly presented, though with some limitations.

The central claim is accurate, where MOMAland is the first standardized benchmark suite for multi-objective multi-agent reinforcement learning. The authors carefully position their work relative to existing frameworks. The empirical demonstrations also support the claim that MOMAland can serve as a useful platform for testing algorithms. The baselines illustrate typical behaviors.

Some claims would benefit from stronger evidence. For instance, while the baselines show feasibility, they are relatively weak and do not convincingly establish the benchmark as a rigorous comparative standard.

**Datasets And Benchmarks:**

The submission gives sufficient detail on how the environments are constructed, how they map to formal MOMARL models, and how they can be accessed and used. The benchmark means the environments and APIs provided through MOMAland rather than some static datasets. They didn't require additional data collected from human subjects.

**Extended Submissions:**

N/A

**Requested Changes:**

A key area for improvement is the set of baseline evaluations. While the paper includes several illustrative baselines, they are intentionally simple and do not reflect the current state of the art in either multi-agent or multi-objective reinforcement learning. Strengthening this component by including more competitive algorithms, or at least situating the provided baselines against known strong performers, would substantially increase the credibility of MOMAland as a fair and useful testbed for algorithm comparison.

Another critical adjustment concerns the statistical robustness of the experiments. Many of the reported results rely on a relatively small number (e.g. 5-10) of random seeds, which limits confidence in the observed trends. Expanding the number of seeds or providing a more thorough justification for the chosen experimental protocol would help ensure that the benchmark results are stable and reproducible.

In addition, while the paper acknowledges that many of the current environments are toy-scale, the implications of this limitation are not fully explored. A more explicit discussion of how these simplifications affect generalizability, and what kinds of conclusions can or cannot be drawn from the environments as currently implemented, would provide readers with more realistic expectations and strengthen the case for adoption.

Finally, the paper would benefit from a clearer benchmarking of computational cost across environments. Some domains, particularly the continuous 3D scenarios, are likely to require substantially more resources than others. Reporting runtime comparisons and resource demands would be highly valuable to practitioners, especially those working with limited compute budgets, and would guide researchers in selecting environments that best suit their experimental constraints.

**Strengths And Weaknesses:**

### Strengths

* First benchmark suite dedicated to multi-objective multi-agent reinforcement learning, filling a clear gap in the ecosystem
* Provides a broad set of more than ten environments covering cooperative, competitive, and mixed-agent settings
* Environments span discrete and continuous states and actions, as well as full and partial observability
* Extends PettingZoo APIs to handle vectorial rewards, supporting both parallel and turn-based interaction modes, offers wrappers for reward normalization, scalarization, and centralization, enabling reuse of existing MARL and MORL algorithms
* Fully integrated with the Farama Foundation ecosystem, ensuring usability and long-term maintenance
* Includes baseline algorithms such as MOMAPPO, Independent Q-Learning, Pareto Conditioned Networks, and GPI-LS
* Demonstrates the benchmark through experiments that highlight meaningful trade-offs and coordination challenges

### Weaknesses

* Most environments are toy-scale or stylized and do not reflect realistic or industrial-scale scenarios
* Some formulations, such as multi-objective Bayesian games, are not yet covered, but this is also mentioned in their limitations
* Few environments provide known optimal Pareto fronts, limiting the ability to benchmark optimality, and the Pareto bound is only from empirical results.
* Baseline algorithms are relatively simple and do not represent state-of-the-art performance
* Experimental analysis is presented environment by environment, with limited cross-environment insights
* Some experiments rely on small numbers of random seeds, which reduces statistical robustness
* The scope is currently limited to foundational testbeds rather than real-world or stochastic domains